# PDZ-directed substrate recruitment is the primary determinant of specific 4E-BP1 dephosphorylation by PP1-Neurabin

Roman O Fedoryshchak[1], Karim El-Bouri[2†], Dhira Joshi[3], Stephane Mouilleron[2], Richard Treisman[1]*

[1]Signalling and Transcription Laboratory, Francis Crick Institute, London, United Kingdom; [2]Structural Biology STP, Francis Crick Institute, London, United Kingdom; [3]Chemical Biology STP, Francis Crick Institute, London, United Kingdom

## eLife Assessment

This **important** study reports on a basis for neurabin-mediated specification of substrate choice by protein phosphatase-1. The data from the comprehensive approach using structural, biochemical, and computational methods are **compelling**. This paper is broadly relevant to those investigating various cellular signaling cascades that entail phosphorylation as the main mechanism.

**\*For correspondence:**
Richard.Treisman@Crick.ac.uk

**Present address:** †Protein Biogenesis Laboratory, Francis Crick Institute, London, United Kingdom

**Competing interest:** The authors declare that no competing interests exist.

**Abstract** Phosphoprotein phosphatase 1 (PP1) relies on association with PP1-interacting proteins (PIPs) to generate substrate-specific PIP/PP1 holoenzymes, but the lack of well-defined substrates has hindered elucidation of the mechanisms involved. We previously demonstrated that the Phactr1 PIP confers sequence specificity on the Phactr1/PP1 holoenzyme by remodelling the PP1 hydrophobic substrate groove. Phactr1 defines a group of 'RVxF-ΦΦ-R-W' PIPs that all interact with PP1 in a similar fashion. Here, we use a PP1-PIP fusion approach to address sequence specificity and identify substrates of the RVxF-ΦΦ-R-W family PIPs. We show that the four Phactr proteins confer identical sequence specificities on their holoenzymes. We identify the 4E-BP and p70 S6K translational regulators as substrates for the Neurabin/Spinophilin PIPs, implicated in neuronal plasticity, pointing to a role for their holoenzymes in mTORC1-dependent translational control. Biochemical and structural experiments show that in contrast to the Phactrs, substrate recruitment and catalytic efficiency of the PP1-Neurabin and PP1-Spinophilin fusions is primarily determined by substrate interaction with the PDZ domain adjoining their RVxF-ΦΦ-R-W motifs, rather than by recognition of the remodelled PP1 hydrophobic groove. Thus, even PIPs that interact with PP1 in a similar manner use different mechanisms to ensure substrate selectivity.

## Introduction

Phosphoprotein phosphatase 1 (PP1) is a member of the PPP superfamily of protein phosphatases, responsible for most cellular serine/threonine dephosphorylation (**Bollen et al., 2010**; **Brautigan and Shenolikar, 2018**). The three PP1 isoforms possess a central metal-binding active centre from which radiate three putative substrate-binding grooves (**Egloff et al., 1995**; **Goldberg et al., 1995**), but have little intrinsic sequence specificity (**Hoermann et al., 2020**). Instead, PP1 substrate specificity and activity are controlled by over 200 PP1-interacting proteins (PIPs), which use a number of short linear motifs (SLIMs) to dock with distinct regions of the PP1 surface, including the substrate-binding grooves (**Cohen, 2002**; **Bollen et al., 2010**; **Casamayor and Ariño, 2020**). PIPs can target PP1 to specific subcellular locations and can control substrate specificity through autonomous

substrate-binding domains, occupation or extension of the substrate grooves, or modification of PP1 surface electrostatics (for references, see *Bollen et al., 2010*; *Fedoryshchak et al., 2020*). Only a few PIP-PP1-substrate interactions are understood in molecular detail, however, and the question of whether PIP interaction imposes sequence specificity on PP1, as opposed to simply bringing enzyme and substrate into close apposition, has remained largely unexplored.

The Phactr family of actin-regulated PP1 cofactors are the only PIPs known to directly impose substrate sequence specificity at the dephosphorylation site itself (*Fedoryshchak et al., 2020*). Like many PIPs, they interact with PP1 using the previously defined 'RVxF', ', and 'R' motifs (for overview, see *Choy et al., 2014*). Structural alignment showed that the Phactrs belong to a small subset of RVxF-ΦΦ-R PIPs whose shared trajectory across the PP1 surface continues to the edge of the hydrophobic groove, making an additional PP1 contact through an additional motif that we termed the 'W' SLIM (*Ragusa et al., 2010*; *Choy et al., 2014*; *Chen et al., 2015*; *Fedoryshchak et al., 2020*; *Yan et al., 2021*; *Figure 1A*, *Figure 1—figure supplement 1*). These 'RVxF-ΦΦ-R-W string' PIPs include Neurabin/Spinophilin (PPP1R9A/B), which play important roles in PP1-dependent regulation of neuronal plasticity (*Allen et al., 1997*; *Burnett et al., 1998*, reviewed by *Sarrouilhe et al., 2006*; *Foley et al., 2021*), PNUTS (PPP1R10), which regulates PolII and chromatin dynamics (*Lee et al., 2010*; *Cortazar et al., 2019*), and PPP1R15A/B, which mediate translational regulation through control of eIF2α dephosphorylation (*Novoa et al., 2001*; *Chen et al., 2015*; *Yan et al., 2021*; *Figure 1A*). In the Phactr1-PP1 holoenzyme, Phactr1 sequences C-terminal to the RVxF-ΦΦ-R-W string interact with the PP1 hydrophobic groove to form a composite hydrophobic pocket, topped by a basic rim. This imposes strong sequence selectivity on substrates, favouring hydrophobic residues at positions +4/+5 relative to the phosphorylated residue, within an acidic context (consensus $pS/T-x_{(2-3)}-\Phi-L$, the 'LLD motif'). Strikingly, this specificity is maintained in a PP1-Phactr1 fusion protein comprising PP1(1–304) linked to Phactr1 sequences from residue 526, just C-terminal to its RVxF motif (*Fedoryshchak et al., 2020*).

Neurabin and Spinophilin remodel the PP1 hydrophobic groove differently from Phactr1, generating a structurally distinct surface on the holoenzyme, and it is likely that the other RVxF-ΦΦ-R-W PIPs do so as well (*Ragusa et al., 2010*; *Fedoryshchak et al., 2020*) (see *Figure 1A*, *Figure 1—figure supplement 2*). Whether and how these diverse surfaces might play a role in the substrate specificity of these PIP/PP1 holoenzymes has remained unclear, largely because little is known about their substrates. Here, we investigate determinants of substrate specificity in the other RVxF-ΦΦ-R-W PIPs, focussing on Neurabin/Spinophilin, which also contain a PDZ domain previously suggested to be involved in substrate recruitment (*Burnett et al., 1998*; *Kelker et al., 2007*). We use the PP1-PIP fusion approach to show that the four Phactr1 holoenzymes have indistinguishable substrate specificities and to identify novel candidate substrates for the other RVxF-ΦΦ-R-W PIP/PP1 complexes. We use 4E-BP1, a new Neurabin/PP1 substrate, to show that unlike the Phactr1/PP1 holoenzyme, substrate specificity of Neurabin/PP1 is largely determined by interaction with the Neurabin PDZ domain rather than the primary sequences of the dephosphorylation site itself.

## Results
### PP1-PIP fusion proteins
The PP1 C-terminal sequences closely approach the Phactr1 RVxF-ΦΦ-R-W string in the vicinity of the ΦΦ motif, which allowed the construction of a single chain PP1α-PIP fusion derivative comprising PP1α(7–304) linked to Phactr1 sequences from a point just C-terminal to its RVxF motif (*Figure 1A*, *Figure 1—figure supplements 1A and 2A*). The remodelled hydrophobic groove of this fusion is structurally identical to that of the Phactr/PP1 holoenzyme, and it retains similar activity and specificity (*Fedoryshchak et al., 2020*). Guided by the structures of the other RVxF-ΦΦ-R-W PIPs, we generated analogous fusions of N-terminally Flag-tagged PP1 with fragments of Neurabin and Spinophilin (*Figure 1—figure supplement 2B and C*) (PPP1R9A/B, *Ragusa et al., 2010*), PNUTS (*Figure 1—figure supplement 2C*), PPP1R10 (*Choy et al., 2014*), and PPP1R15A and PPP1R15B (*Figure 1—figure supplement 2E and F*; *Chen et al., 2015*; *Yan et al., 2021*), comprising the PP1-interacting sequences and any known protein interaction domains immediately C-terminal to them. We also constructed fusions with each of the four Phactr proteins to explore any variation in their sequence specificities (*Figure 1A*; *Figure 1—figure supplements 1A and 2A*).

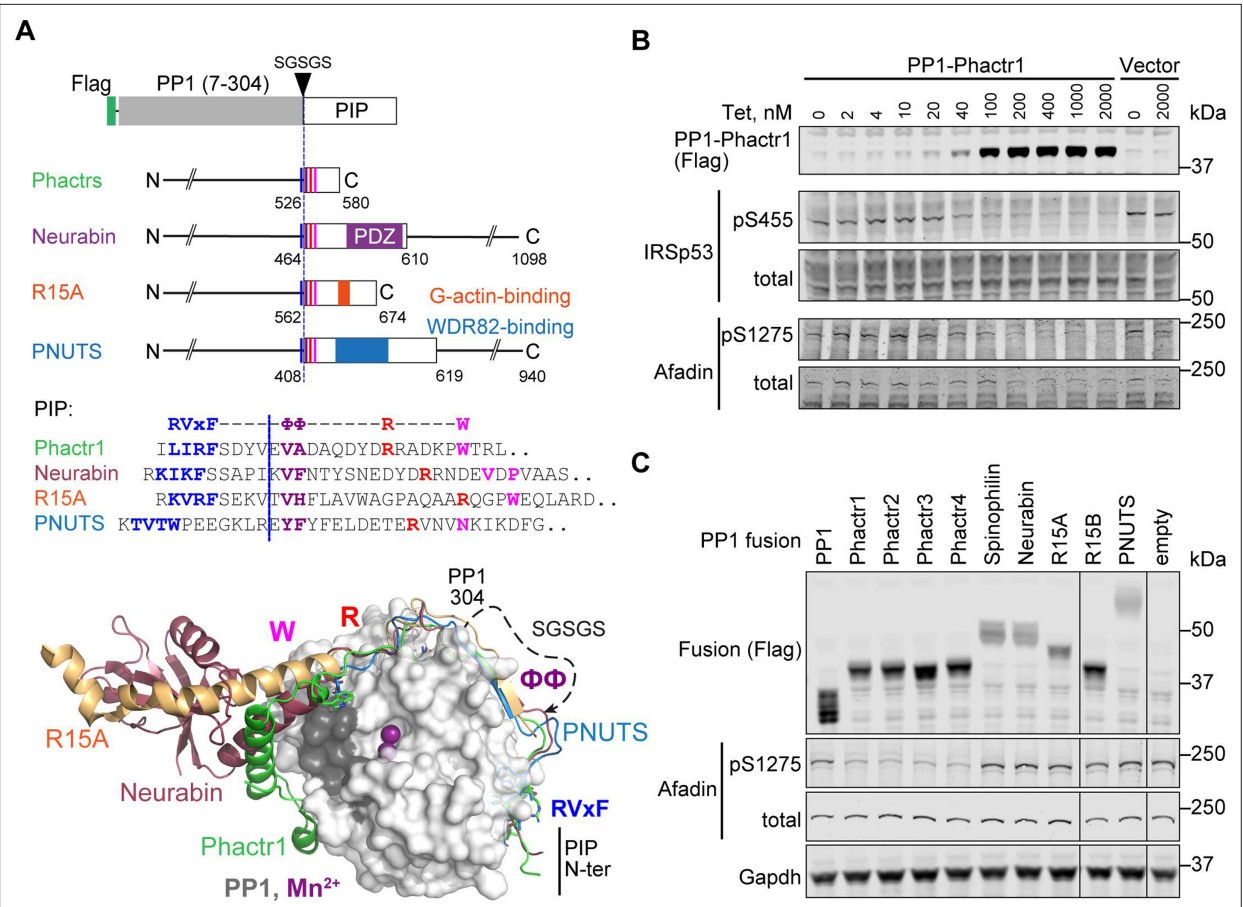

**Figure 1.** PP1-PIP fusion proteins. (**A**) Structures of fusion proteins. N-terminally Flag-tagged PP1α(7–304) is linked to sequences from each of the four families of RVxF-ΦΦ-R-W PIPs, shown as an open box. Each fusion contains sequences immediately C-terminal to the PP1 interaction motif (coloured lines), including known protein interaction domains previously implicated in potential substrate interactions (coloured blocks). For PIP sequences in each fusion, see *Figure 1—figure supplement 1A* and Methods. Middle, sequences of the RVxF-ΦΦ-R-W string in each PIP, with motifs coloured. Each fusion contains the sequences C-terminal to the dashed line, representing the position of PP1-SGSGS linker insertion. Bottom, structures of PP1/PIP complexes. Crystal structures of different PIP/PP1 complexes superimposed, aligned on PP1. Grey: PP1 (PDB: 4MOV), with PIP sequences as follows; green, Phactr1 (PDB: 6ZEE); magenta, Neurabin (PDB: 3HVQ); orange, R15A (PDB: 7NZM); blue, PNUTS (PDB: 4MOY). Dashed line, GSGSG linker. (**B**) Activity of PP1-Phactr1 expressed in Flp-In T-REx 293 cells. PP1-Phactr1 expression was induced by tetracycline as indicated. Phosphorylation of Phactr1/PP1 substrates IRSp53 S455 and Afadin S1275 is shown below. (**C**) Analysis of Phactr1/PP1 substrate Afadin pS1275 phosphorylation in Flp-In T-REx 293 cells expressing PP1 and PP1-fusion proteins.

The online version of this article includes the following source data and figure supplement(s) for figure 1:

**Source data 1.** Sequence alignments related to *Figure 1A*, *Figure 1—figure supplement 1A*.

**Source data 2.** Original files for western blot analysis displayed in *Figure 1B*.

**Source data 3.** Full-size western blots indicating the relevant bands and treatments related to *Figure 1B*.

**Source data 4.** Original files for western blot analysis displayed in *Figure 1C*.

**Source data 5.** Full-size western blots indicating the relevant bands and treatments related to *Figure 1C*.

**Figure supplement 1.** Sequences and functional validation of PP1-PIP fusions.

**Figure supplement 1—source data 1.** Original files for western blot analysis displayed in *Figure 1—figure supplement 1B*.

**Figure supplement 1—source data 2.** Full-size western blots indicating the relevant bands and treatments related to *Figure 1—figure supplement 1B*.

**Figure supplement 2.** Structural alignments of RVxF-ΦΦ-R-W PIP complexes with PP1.

Each fusion protein was stably expressed in 293 Flp-In T-REx cells using a tetracycline-inducible vector (*Ward et al., 2011*). Tetracycline titration of PP1-Phactr1 cells induced increasing expression of the fusion protein, leading to corresponding dephosphorylation of its substrates IRSp53 pS455 and Afadin pS1275 (*Fedoryshchak et al., 2020*; *Figure 1B*). Afadin pS1275 was dephosphorylated by all four PP1-Phactr fusions, but not by the other PP1-PIP fusions (*Figure 1C*). Similar results were observed with exogenously expressed wildtype IRSp53. In this setting, alanine substitution of IRSp53 L460, which contacts the novel Phactr1/PP1 hydrophobic pocket and impairs dephosphorylation by the intact Phactr1/PP1 holoenzyme (*Fedoryshchak et al., 2020*), also impaired dephosphorylation by all the PP1-Phactr fusions, indicating that they recognise phosphorylated IRSp53 in a similar way (*Figure 1—figure supplement 1B*).

## Proteomic analysis of PP1-fusion specificity

To investigate the substrate specificities of the fusion proteins, we performed tandem mass tag (TMT) phosphoproteomics. Fractionated peptides were measured in both MS2 and MS3 modes for maximal identification and quantification (*Figure 2A*). Phosphorylation site abundances within triplicate samples from the same cell line were comparable between replicates (*Figure 2B*). First, we identified phosphorylation sites significantly depleted by expression of PP1α(7–304) alone compared with vector only, using Perseus software and a t-test with a 1% permutation-based false discovery rate cutoff (*Tyanova et al., 2016*). In agreement with previous results (*Hoermann et al., 2020*), PP1 exhibited little sequence specificity, other than a preference for positively charged residues in positions −3, −1, and +3 to the dephosphorylation site (*Figure 2—figure supplement 1A and B*).

We then compared the specificities of the four PP1-Phactr fusion proteins. Expression of PP1-Phactr1 revealed numerous phosphorylation sites that were specifically depleted compared with cells expressing PP1α alone (*Figure 2C*). This population contained many of the Phactr1/PP1 holoenzyme substrates previously identified in neurons or NIH3T3 fibroblasts, including IRSp53 pS455 and Afadin pS1275 (*Figure 2C*; *Fedoryshchak et al., 2020*). Of the 28 top Phactr1/PP1 hits previously identified in NIH3T3 cells, 18 were detectable in the 293 system, of which 13 also registered as PP1-Phactr1 hits (*Figure 2—figure supplement 1C*). The putative PP1-Phactr1 substrates were also substantially enriched in the Phactr1/PP1 consensus dephosphorylation motif S/T-$x_{2,3}$-$\Phi$-L (*Figure 2D*). The profiles of PP1-Phactr2, PP1-Phactr3, and PP1-Phactr4 cells were substantially similar to that of PP1-Phactr1 (*Figure 2—figure supplement 1D–F*), exhibiting good overall correlations both between total phosphorylation site profiles (*Figure 2B*), and significant enrichment for both the S/T-$x_{2,3}$-$\Phi$-L motif and specific Phactr1/PP1 substrates (*Figure 2D*; *Figure 2—figure supplement 1C*); over 60% of the depleted phosphorylation sites were in common between all four fusions (*Figure 2E*). These data show that the PP1-Phactr fusions recapitulate the specificity of the Phactr1/PP1 holoenzyme.

We next used the proteomics approach to investigate protein dephosphorylation by the PP1-R15A/B and PP1-PNUTS fusion proteins, comparing each fusion to all the others (other than its paralog) or PP1α alone (*Figure 2B*). PPP1R15A has a well-validated substrate, eIF2α pS51 (*Novoa et al., 2001*), whose effective dephosphorylation requires additional recruitment of G-actin to the PPP1R15A/PP1 complex (*Chen et al., 2015*; *Yan et al., 2021*). The eIF2α pS51 phosphorylation site was not detected in the dataset, however, and no other phosphorylation sites were detectably depleted, apart from PPP6R1 S531, in cells expressing PPP1R15B (*Figure 2—figure supplement 2A and B*). Expression of the PP1-PNUTS fusion, which includes sequences that recruit WDR82 (*Lee et al., 2010*), led to specific depletion of phosphorylation sites from CXXC1/CFP1 and SET1B, which along with WDR82 are components of the COMPASS histone lysine *N*-methyl transferase complexes (*Cenik and Shilatifard, 2021*; *Figure 2—figure supplement 2C*). These proteins bear no obvious sequence similarity in the vicinity of the dephosphorylation site (*Figure 2—figure supplement 2D*). These findings will be explored in future work.

## New candidate substrates for Neurabin and Spinophilin

Having validated the PP1-PIP fusion approach, we focussed on Neurabin and Spinophilin, two PIPs implicated in neuronal plasticity (*Sarrouilhe et al., 2006*; *Foley et al., 2021*), which contain a PDZ domain implicated in recruitment of potential substrates (*Burnett et al., 1998*; *Yan et al., 1999*; *Sarrouilhe et al., 2006*; *Kelker et al., 2007*). The PDZ domain is separated from the RVxF-$\Phi\Phi$-R-W string by a 5-turn α-helix, which remodels the PP1 hydrophobic groove in a manner distinct from

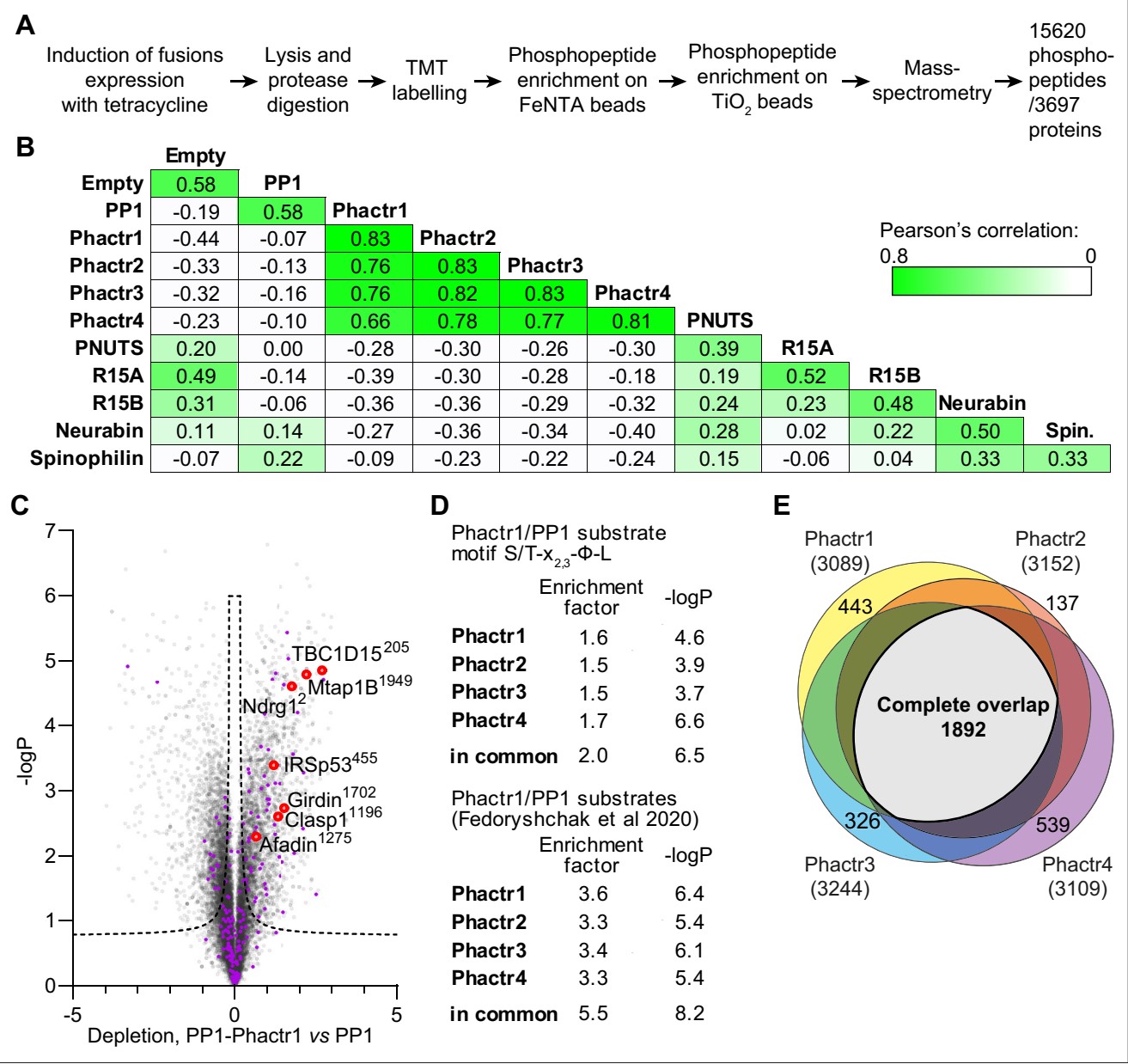

**Figure 2.** PP1-PIP fusion phosphoproteomics. (**A**) Tandem mass tag (TMT) phosphoproteomics workflow. (**B**) Average sample-to-sample correlations between triplicates from cells expressing the different fusion proteins, PP1α(7–304)-SGSGS alone, or empty vector. For the same fusion-expressing cell lines, the average of Pearson coefficients of correlation within a triplicate are shown. (**C**) Specific phosphosite depletion in cells expressing PP1-Phactr1 as opposed to PP1 alone. Abundances of specific phosphosites in PP1 and PP1-Phactr1 samples were determined, log-transformed, and expressed as Z-scores. For each phosphosite, depletion in cells expressing PP1-Phactr1 as opposed to PP1 alone was quantified as the difference between the PP1 and PP1-Phactr1 Z-scores, and plotted versus $-\log_{10}p$. Dashed line, 5% false discovery rate cut-off. Purple, phosphosites conforming to the Phactr1/PP1 substrate motif S/T-$x_{2,3}$-Φ-L. Red, Phactr1/PP1 substrates identified previously (*Fedoryshchak et al., 2020*). (**D**) Enrichment of hits conforming to the Phactr1 substrate motif S/T-$x_{2,3}$-Φ-L and of hits identified in the previous study in all Phactr samples calculated using Fisher's exact test. (**E**) Venn diagram showing overlap between hits identified as potential Phactr1-4 substrates.

The online version of this article includes the following figure supplement(s) for figure 2:

**Figure supplement 1.** Detailed analysis of PP1 and PP1-Phactr1-4 phosphoproteomics.

**Figure supplement 2.** Detailed analysis of PP1-R15A, PP1-R15B and PP1-PNUTS phosphoproteomics.

Phactr1 (*Figure 1A*, *Figure 1—figure supplement 2G*; *Ragusa et al., 2010*; *Fedoryshchak et al., 2020*).

Expression of the PP1-Neurabin or PP1-Spinophilin fusions led to specific depletion of closely related sets of phosphorylation sites (*Figure 3A and B*). These included multiple sites from the translational inhibitor proteins 4E-BP1 and 4E-BP2. Levels of total 4E-BP1 and 4E-BP2 proteins were not affected (*Figure 3—figure supplement 1A*). Phosphorylation sites from two other proteins, DTL and CAMSAP3, were also significantly depleted upon expression of both PP1-Spinophilin and PP1-Neurabin fusions, while a further 7 were specific to one fusion or the other (see Discussion). However, apart from a preference for proline at position +1, inspection of these sequences did not reveal any obvious sequence similarities in the vicinity of the dephosphorylation site (*Figure 3C*). Substrates identified in the PP1-Phactr and PP1-PNUTS screens showed no detectable depletion, and vice versa, indicating that the substrates identified were specific for each fusion. Consistent with the phosphoproteomics data, immunoblotting analysis with phosphorylation-specific antibodies demonstrated that induction of PP1-Neurabin resulted in decreased phosphorylation of T70, S65/S101, and possibly T37/T46, while total 4E-BP1 resolved from a heterogeneous distribution to a largely monodisperse species (*Figure 3D*, *Figure 3—figure supplement 1B and C*).

The 4E-BPs are critical components of the mTORC1 growth control pathway which couples translation to nutrient availability and extracellular signals (reviewed by *Hoeffer and Klann, 2010*; *Liu and Sabatini, 2020*) (see Discussion). Phosphorylation of 4E-BPs potentiates translation by inhibiting their ability to sequester EIF4E (reviewed by *Martineau et al., 2013*; *Romagnoli et al., 2021*). Accordingly, expression of PP1-Neurabin, but not PP1 alone, suppressed translation in 293 cells (*Figure 3E*). These data establish the Neurabin/PP1 and Spinophilin/PP1 holoenzymes as potential negative regulators of the mTORC1 pathway (*Figure 3—figure supplement 1D*; see Discussion).

## The PP1-Neurabin PDZ domain is required for 4E-BP1 dephosphorylation

To demonstrate a direct enzyme-substrate relationship between 4E-BP1 and PP1-Neurabin, we expressed mCherry-tagged 4E-BP1 in 293 cells, recovered it on RFP-trap affinity beads, and incubated it with increasing amounts of recombinant PP1-Neurabin. Analysis with the phospho-specific antibodies confirmed that pT37/46, pS65/101, and pT70 are all direct targets for dephosphorylation by PP1-Neurabin, with pT70 being somewhat preferred (*Figure 4A and B*). The 4E-BP C-terminal sequences are similar to those of p70 S6K, which was previously shown to interact with the Neurabin PDZ domain through a C-terminal PDZ-binding motif (PBM) (*Burnett et al., 1998*; *Figure 4C*). We therefore considered the possibility that 4E-BP1 dephosphorylation at multiple sites by PP1-Neurabin reflects its recruitment through PDZ interaction.

Simple deletion of the 4E-BP1 C-terminal residues substantially blocked phosphorylation of transfected mCherry-4E-BP1 (*Figure 4—figure supplement 1A*), reflecting the loss of the C-terminal TOR signalling (TOS) motif required for mTORC1 kinase recruitment and 4E-BP1 phosphorylation (*Schalm and Blenis, 2002*; *Yang et al., 2017*). Noting that the TOS motif does not include the C-terminal carboxylate, an essential part of the classical PBM (*Harris and Lim, 2001*; *Tonikian et al., 2008*; *Subbaiah et al., 2011*), we generated mCherry-4E-BP1(118+A): this contains an additional C-terminal alanine, which leaves the TOS motif intact but inactivates the PBM. Indeed, although mCherry-4E-BP1(118+A) was efficiently phosphorylated upon expression in 293 cells (*Figure 4—figure supplement 1A*), its dephosphorylation in vitro required PP1-Neurabin concentrations some 20–50 times greater than the wildtype protein (*Figure 4A and B*). These results suggest that the 4E-BP1 C-terminal sequences constitute a PBM and point to a role for the Neurabin and Spinophilin PDZ domains in substrate recognition.

We next used a fluorescence polarisation (FP) assay to compare the PDZ-binding affinity of the 4E-BP C-terminal sequences with those of p70 S6K and other proteins reported to be Neurabin/Spinophilin PDZ ligands (*Figure 4C*; *Burnett et al., 1998*; *Penzes et al., 2001*; *Kelker et al., 2007*; *Ragusa et al., 2010*). The 4E-BP1 PBM, which is identical amongst all three 4E-BP isoforms, bound the PDZ domains with comparable affinities in the micromolar range, and binding was abolished by substitution of the C-terminal hydrophobic residues (*Figure 4C and D*; *Figure 4—figure supplement 1B*). These binding affinities were 10- to 30-fold greater than those of p70 S6K and the RhoGEF Kalirin-7, which were of order 100 μM, and >100-fold greater than various glutamate receptor C-terminal

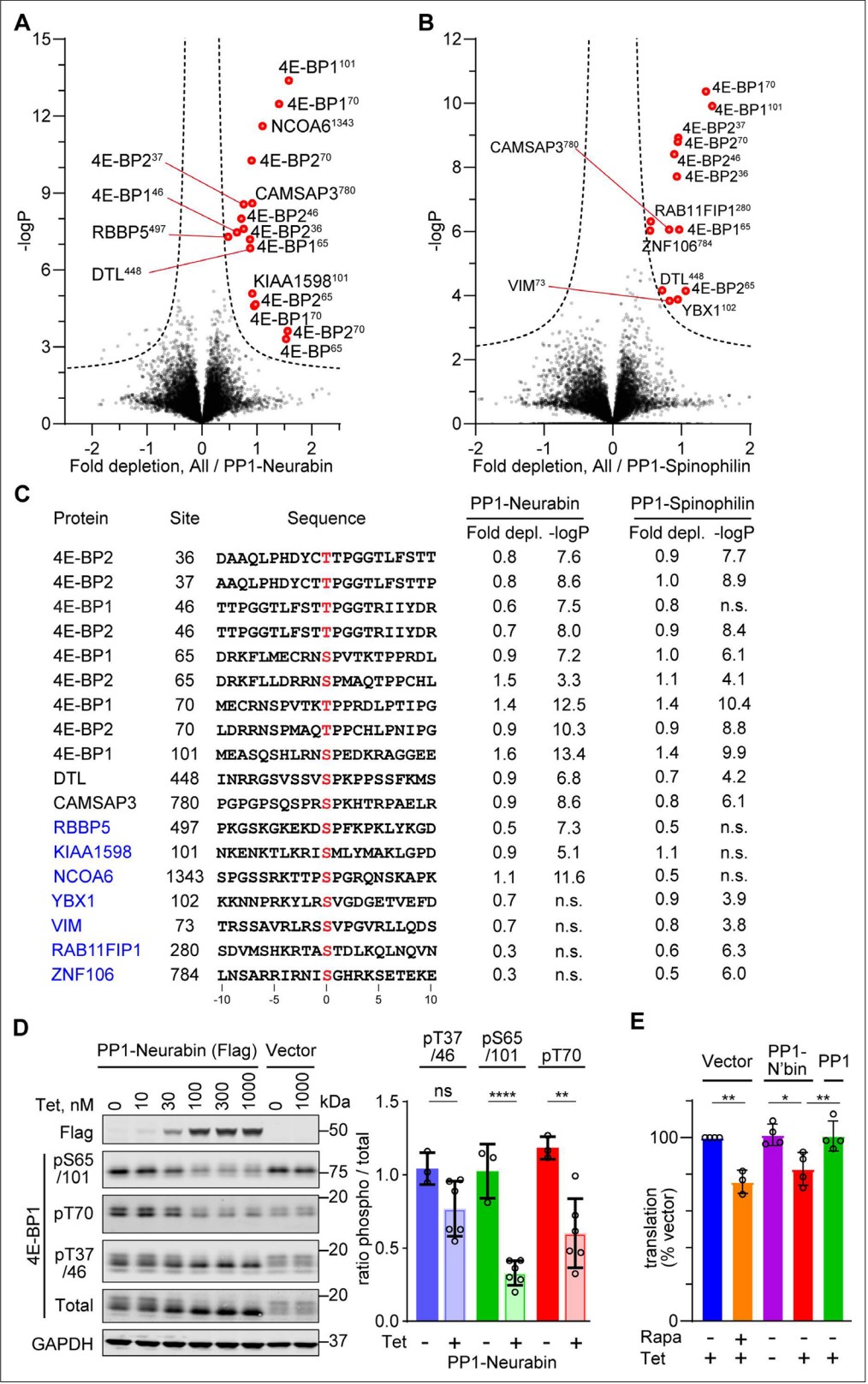

**Figure 3.** Phosphoproteomics of PP1-Neurabin and PP1-Spinophilin. (**A**) Identification of PP1-Neurabin substrates. Abundances of specific phosphorylation sites in PP1-Neurabin samples were determined, log-transformed, and expressed as Z-scores. For each phosphosite, the abundance in the remaining datasets, excluding PP1-Spinophilin, was quantified in the same way. Depletion of phosphosites in cells expressing PP1-Neurabin was quantified as

*Figure 3 continued*

the difference between the PP1-Neurabin and the dataset average Z-scores, and plotted versus -log$_{10}$p. Dashed line, 5% false discovery threshold; significantly depleted phosphosites are highlighted in red. (**B**) Identification of PP1-Spinophilin substrates. Depletion of each phosphorylation site in cells expressing PP1-Spinophilin, relative to its average abundance in the other datasets, excluding PP1-Neurabin, was quantified and plotted as in (**A**). (**C**) Sequences of significantly depleted phosphorylation sites identified in (**A** and **B**). (**D**) Immunoblot analysis of 4E-BP1 phosphorylation sites in 293 Flp-In T-REx cells upon induced expression of PP1-Neurabin or empty vector. Note that the low level of PP1-Neurabin expression in uninduced cells (see *Figure 3—figure supplement 1C*) alters the relative abundance of the different phosphorylated forms compared with 293 Flp-In T-REx cells expressing vector alone. (**E**) Protein synthesis quantification assay. 293 Flp-In T-REx cells expressing vector alone, PP1-Neurabin, or PP1, were induced with tetracycline (50 nM) and/or treated with rapamycin (50 nM) for 16 hr as indicated before treatment with *O*-propargyl puromycin to label nascent polypeptides, which were conjugated to Alexa Fluor-488 azide and quantified by flow cytometry. Fluorescence intensities were normalised to untreated cells.

The online version of this article includes the following source data and figure supplement(s) for figure 3:

**Source data 1.** Original files for western blot analysis displayed in *Figure 3D*, *Figure 4—figure supplement 1C*.

**Source data 2.** Full-size western blots indicating the relevant bands and treatments related to *Figure 3D*, *Figure 4—figure supplement 1C*.

**Source data 3.** Flow cytometry data related to *Figure 3E*.

**Figure supplement 1.** Additional details for the identification of 4E-BP1 as PP1-Neurabin target.

**Figure supplement 1—source data 1.** Original files for western blot analysis displayed in *Figure 3—figure supplement 1B*, *Figure 4—figure supplement 1A*.

**Figure supplement 1—source data 2.** Full-size western blots indicating the relevant bands and treatments related to *Figure 3—figure supplement 1B*, *Figure 4—figure supplement 1A*.

---

peptides (*Figure 4C and D*; *Figure 4—figure supplement 1B*). p70 S6K also functions in the mTORC1 pathway (for reviews, see *Liu and Sabatini, 2020*; *Artemenko et al., 2022*). Although most of its phosphorylation sites were not detected by phosphoproteomics, immunoblotting experiments demonstrated that PP1-Neurabin expression decreased the levels of the activating T389 phosphorylation (*Figure 4—figure supplement 1C*).

## PDZ domain interaction determines PP1-Neurabin specificity

We next compared how interactions with the PP1-Neurabin PDZ and the remodelled PP1 hydrophobic groove contribute to substrate specificity. To do this, we compared the ability of PP1-Neurabin and PP1-Phactr1 to dephosphorylate synthetic peptides derived from their substrates 4E-BP1 and IRSp53. In these peptides, 14$^{mer}$ sequences spanning 4E-BP1 and IRSp53 dephosphorylation sites are joined via a GSG linker to wildtype or mutant 4E-BP1 PBM sequences (*Figure 5A*). As a representative PP1-Neurabin substrate, we used 4E-BP1 residues 64–78, including the phosphorylated T70 site, linked to intact or mutant 4E-BP1 PBM (substrates 4E-BP1$^{PBM}$ and 4E-BP1$^{MUT}$). For comparison, we used Phactr1 substrate peptides comprising IRSp53 residues 449–463, spanning the phosphorylated S455 site, also linked to the intact or mutant 4E-BP1 PBM (substrates IRSp53$^{PBM}$ and IRSp53$^{MUT}$; *Figure 5A*, *Figure 5—figure supplement 1A*). These substrates, or their mutated derivatives, were then dephosphorylated using PP1-Neurabin, PP1-Phactr1, or PP1 alone. Results are summarised in *Figure 5*, *Figure 5—figure supplement 1*; the assay data and statistical analysis provided in *Supplementary file 3*.

PP1-Neurabin dephosphorylated 4E-BP1$^{PBM}$ with a catalytic efficiency some 100-fold greater than PP1 alone, while peptide 4E-BP1$^{MUT}$, which cannot bind the Neurabin PDZ domain, was 30-fold less reactive (*Figure 5B*, *Figure 5—figure supplement 1A*). In contrast, PP1-Phactr1 dephosphorylated 4E-BP1$^{PBM}$ and 4E-BP1$^{MUT}$ at rates similar to those achieved by PP1 alone (*Figure 5B*, *Figure 5—figure supplement 1A*). The Neurabin sequences, specifically the PDZ domain, thus play a critical role in specific 4E-BP1 pT70 substrate recognition. PP1-Neurabin also dephosphorylated IRSp53$^{PBM}$, which contains the 4E-BP1 PBM, with somewhat higher catalytic efficiency to that seen with 4E-BP1$^{PBM}$ itself (*Figure 5B*, *Figure 5—figure supplement 1A and B*). This was entirely dependent on PDZ domain interaction, IRSp53$^{MUT}$ being ~100-fold less reactive (*Figure 5B*, *Figure 5—figure supplement 1A and B*). In contrast, PP1-Phactr1 dephosphorylated both IRSp53$^{PBM}$ and IRSp53$^{MUT}$ peptides

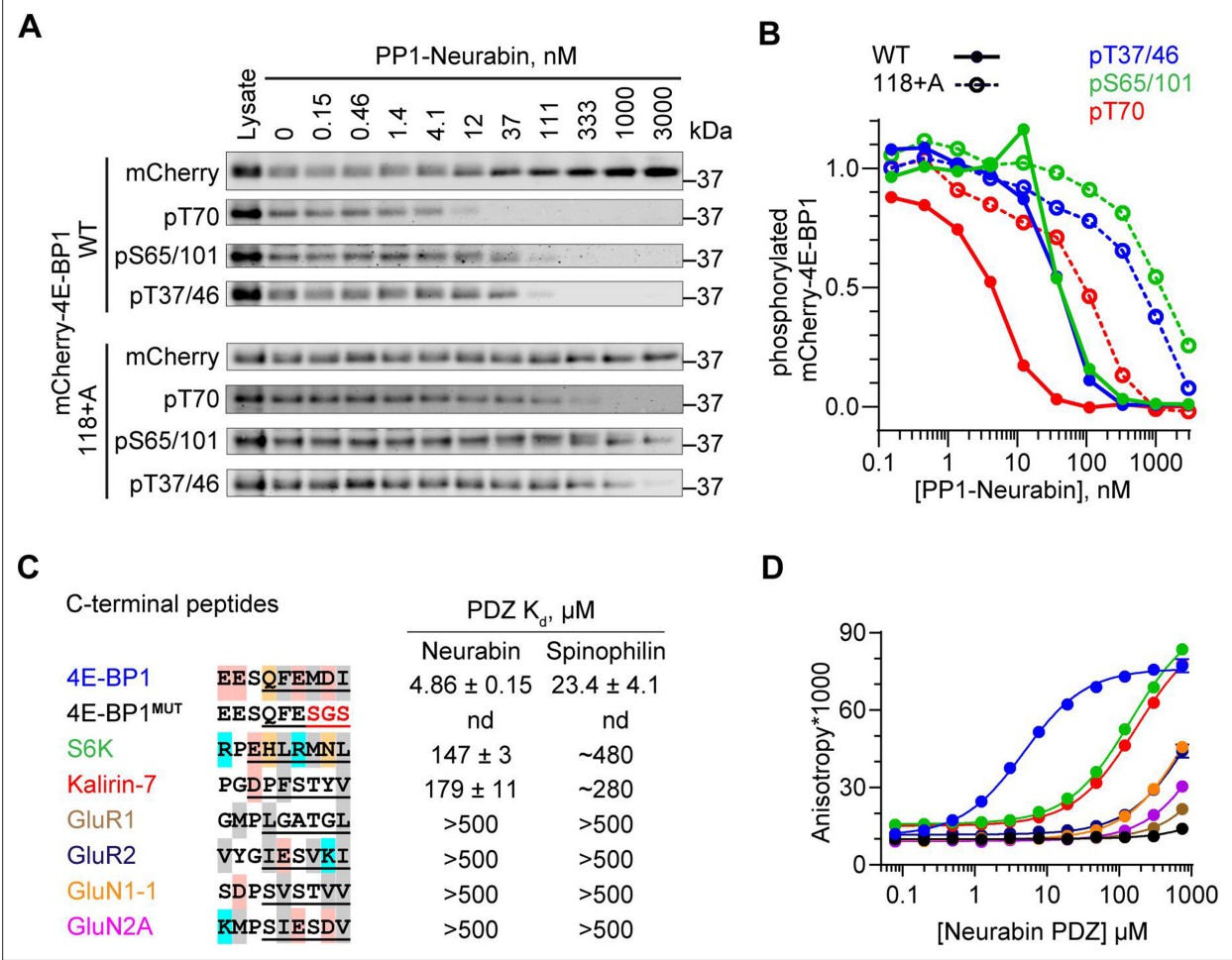

**Figure 4.** 4E-BP1 is a substrate of PP1-Neurabin. (**A**) mCherry-tagged wildtype 4E-BP1 or 4E-BP1(118+A) were expressed and purified from 293 cells, incubated with increasing amounts of recombinant PP1-Neurabin. Phosphorylation of the indicated sites was analysed by immunoblotting. (**B**) Quantification of (**A**). (**C**) Left, sequence alignment of potential Neurabin/Spinophilin PDZ domain ligands. Grey shading, hydrophobic residues; pink, acidic residues; cyan, basic residues; orange, hydrophilic residues. Underlining shows sequences N-terminally linked to 6-carboxyfluorescein (FAM) for use in fluorescence polarisation (FP) assay. Right, binding affinities for the Neurabin and Spinophilin PDZ domains as determined in the FP assay. (**D**) FP assay. FAM-labelled peptides (see C) were titrated with increasing concentrations of recombinant Neurabin PDZ domain and affinity estimated from change in fluorescence anisotropy. For Spinophilin data, see *Figure 4—figure supplement 1B*.

The online version of this article includes the following source data and figure supplement(s) for figure 4:

**Source data 1.** Original files for western blot analysis displayed in *Figure 4A*.

**Source data 2.** Full-size western blots indicating the relevant bands and treatments related to *Figure 4A*.

**Source data 3.** Fluorescence polarisation assay related to *Figure 4C, D*, *Figure 4—figure supplement 1A, B*.

**Figure supplement 1.** 4E-BP1 is a substrate of both PP1-Neurabin and PP1-Spinophilin.

with a similar catalytic efficiency, some 100-fold greater than that seen with PP1 alone (*Figure 5B*, *Figure 5—figure supplement 1A and B*). These results demonstrate the critical role played by the PDZ domain substrate specificity and underscore the role played by PIPs in potentiating the catalytic efficiency of PIP/PP1 complexes.

## Substrate interactions with the remodelled PP1 hydrophobic groove do not affect PP1-Neurabin specificity

We next assessed the potential role of the remodelled hydrophobic groove in substrate recognition. Positions +3 to +6 relative to the dephosphorylation site are critical for recognition of the remodelled hydrophobic groove by Phactr1/PP1 (*Fedoryshchak et al., 2020*), so we assessed the effect

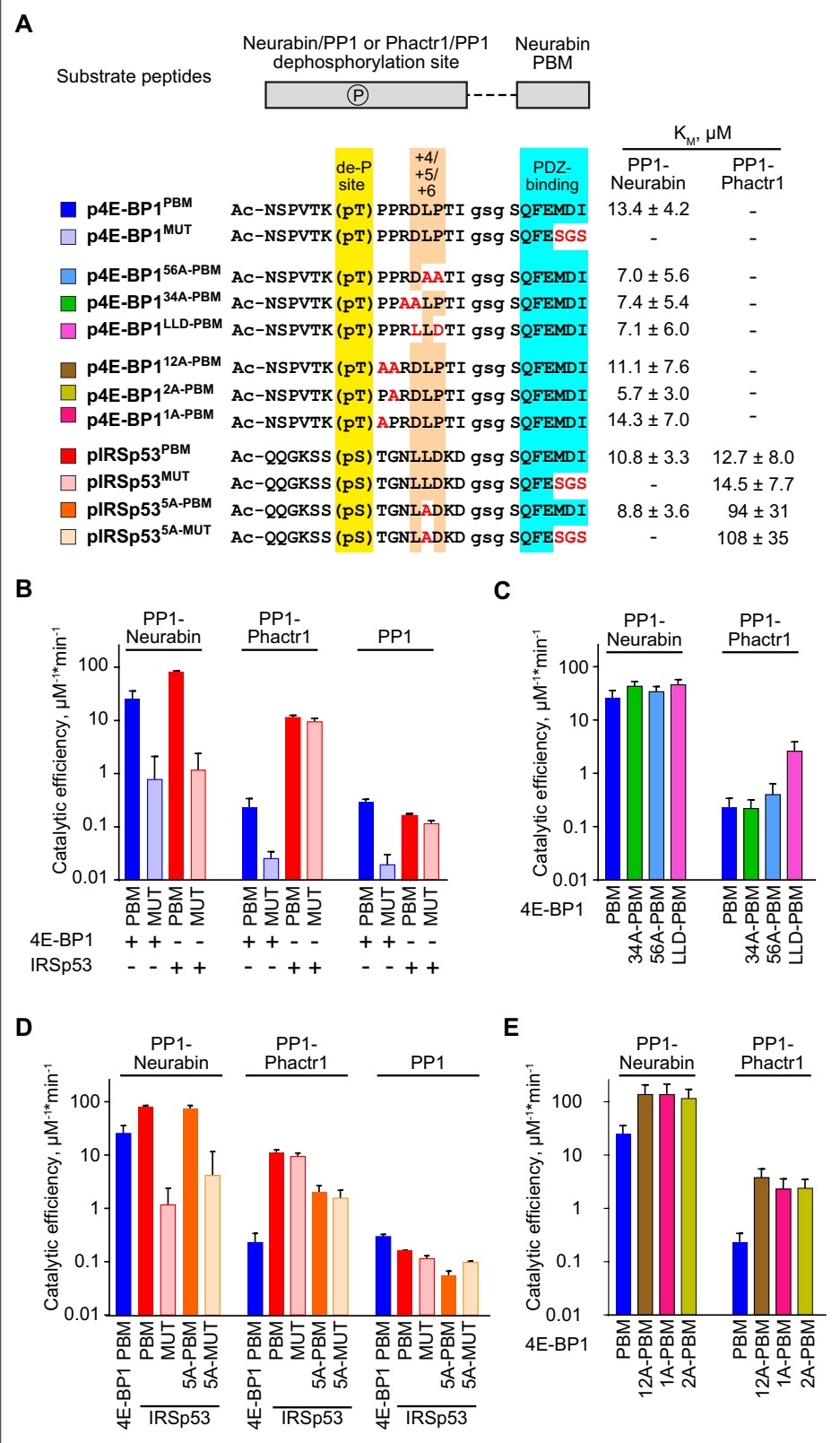

**Figure 5.** Substrate specificity determinants of PP1-Neurabin. (**A**) Top, synthetic substrate peptides contain either the 4E-BP1 T70 or IRSp53 S455 phosphorylation sites, joined by a GSG linker to the Neurabin PDZ-binding C-terminal sequences. PBM, PDZ-binding motif (FEMDI); MUT, mutated PBM (FE**sgs**). Below, sequences of the different peptides analysed; highlights indicate the dephosphorylation site (yellow), the +4/+6 region (orange),

*Figure 5 continued on next page*

*Figure 5 continued*

and the PDZ-binding sequence (cyan), with alanine and other substitutions indicated in red. Peptides were treated with recombinant PP1-Neurabin, PP1-Phactr1, or PP1 in the presence of the phosphate sensor, and $K_M$ and catalytic efficiencies determined. $K_M$ are shown at the right; for catalytic efficiency quantification, see *Figure 5—figure supplement 1A*. For raw and processed data, see *Supplementary file 3*. (**B–E**) Panels show relative catalytic efficiencies as determined from data displayed in *Figure 5—figure supplement 1B–E*. Each panel shows different subsets of the data to highlight comparison between different enzymes and/or substrates. For raw and processed data, see *Supplementary file 3*. (**B**) Comparison of Neurabin-PP1 and Phactr1-PP1 substrates 4E-BP1 and IRSp53 to assess the role of the Neurabin PDZ domain in substrate recognition. (**C**) Role of the +4/+6 region in 4E-BP1 substrate recognition. (**D**) Role of the +5 residue in IRSp53 substrate recognition. (**E**) Role of 4E-BP1+1/+2 residues.

The online version of this article includes the following source data and figure supplement(s) for figure 5:

**Source data 1.** Activity assay data related to *Figure 5*, *Figure 5—figure supplement 1*.

**Figure supplement 1.** Substrate dephosphorylation by PP1-Neurabin, PP1-Phactr1 and PP1.

**Figure supplement 2.** Schematic showing the mechanisms of substrate binding by Phactr1/PP1 and Neurabin/PP1 complexes.

of mutations at these positions on catalytic activity. PP1-Neurabin dephosphorylated 4E-BP1^PBM containing alanine substitutions, or IRSp53 sequences, at positions +3/4 or +5/6 with slightly increased catalytic efficiency (*Figure 5C*, *Figure 5—figure supplement 1A and C*). These mutations had strikingly different effects on 4E-BP1^PBM dephosphorylation by PP1-Phactr1, however. While the alanine substitutions at positions +3/4 or +5/6 had little effect, conversion of +4 to +6 to the IRSp53 sequence LLD increased catalytic efficiency some 20-fold (*Figure 5C*, *Figure 5—figure supplement 1A and C*). Similar results were seen with the IRSp53 substrate peptides: alanine substitution in the groove-interacting region had no effect on catalytic efficiency with PP1-Neurabin, but significantly impaired it with PP1-Phactr1 (*Figure 5D*, *Figure 5—figure supplement 1A and D*). Strikingly, alanine substitutions at +1 and+2 in 4E-BP1^PBM significantly increased catalytic efficiency by both PP1-Neurabin and PP1-Phactr1, perhaps reflecting changes at the catalytic site itself (*Figure 5E*, *Figure 5—figure supplement 1A and E*; see Discussion).

Taken together with the results in the preceding section, these data support a model in which PP1-Neurabin substrate specificity is driven predominantly by the ability of substrates to interact with the Neurabin PDZ domain rather than the remodelled PP1 hydrophobic groove (*Figure 5—figure supplement 2*). In contrast, the specificity of PP1-Phactr1, like that of the PP1/Phactr1 holoenzyme, is critically dependent on interaction with the remodelled PP1 hydrophobic groove (*Fedoryshchak et al., 2020*; *Figure 5—figure supplement 2*; see Discussion).

## Structural analysis of PP1/Neurabin-4E-BP1 interaction

We next sought to visualise PP1-Neurabin/4E-BP1 interactions directly at the structural level. To do this, we used the 'chimera' strategy previously used to examine Phactr1/PP1/substrate interactions, in which Phactr1 substrate sequences were fused C-terminally to PP1(7–304), and co-crystallised with the Phactr1 PP1-interacting C-terminal domain. This revealed a putative enzyme-product complex, with substrate sequences binding in a remodelled PP1 hydrophobic groove and serine and a presumed phosphate docked at the active site (*Fedoryshchak et al., 2020*). Accordingly, we constructed an analogous PP1-4E-BP1 substrate fusion, comprising PP1(7–304)/(SG)₅/4E-BP1(65–83)/G/4E-BP1(112–118) (*Figure 6A*), co-expressed it with Neurabin residues 423–593, and determined the crystal structure of the purified complex at 2.36 Å resolution (*Figure 6B*, *Figure 6—figure supplement 1*; *Table 1*; *Ragusa et al., 2010*).

In the complex, the PP1 catalytic core and Neurabin PDZ sequences were well resolved, along with the 4E-BP1 PBM, which was docked with the Neurabin PDZ domain (*Figure 6B*). The individual domains were largely identical to those determined previously for unliganded Neurabin/PP1 (*Ragusa et al., 2010*) (PP1(1–304), RMSD 0.22 Å over 269 Cα, PDZ(501–592) RMSD 0.39 Å over 65 Cα). However, the PDZ domain in our Neurabin/PP1-4E-BP1 substrate complex structure is oriented at 22° to that in the unliganded Neurabin/PP1 complex, reflecting a slight bend in the C-terminal section of the 5-turn α-helix that connects it to the RVxF-Φ Φ-R-W string (*Figure 6C*). The 4E-BP1 PBM, 113(QFEMDI)118, uses a beta-strand addition mechanism to make extensive contacts with the Neurabin PDZ domain, similar to those seen in other PDZ-ligand complexes, that widen the PBM-binding

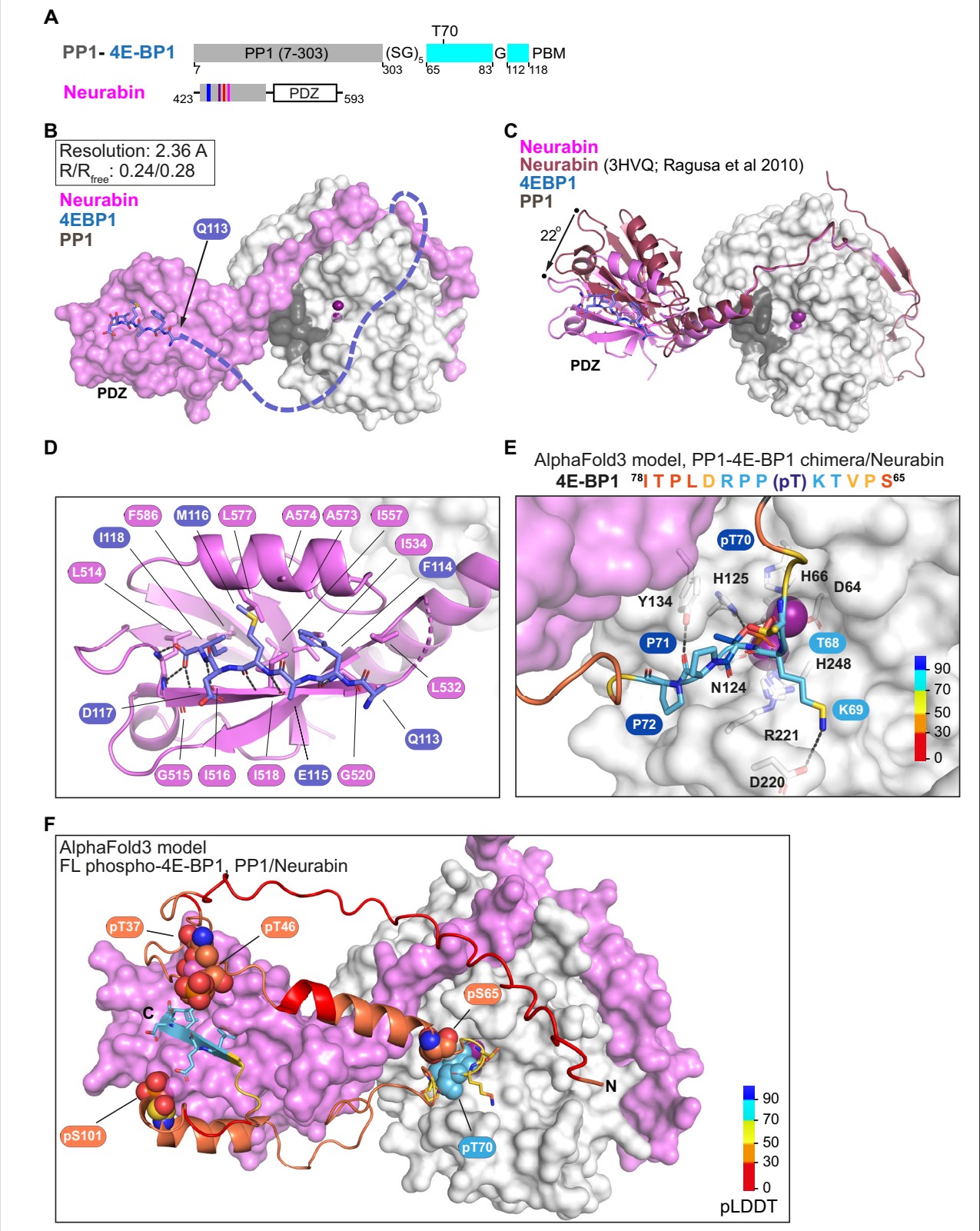

**Figure 6.** Structural analysis of 4E-BP1 interactions with PP1-Neurabin. (**A**) Schematic of the PP1-4E-BP1 chimera and of Neurabin PP1-interacting and PDZ domain sequences. (**B**) Crystal structure of the PP1-4E-BP1/Neurabin complex. PP1 in white surface representation, Neurabin in lilac surface representation, 4E-BP1 in blue stick representation, with unresolved sequences indicated by dashed line. PP1 active site presumptive Mn²⁺ ions in purple. (**C**) Comparison of PP1-4E-BP1/Neurabin complex structure with the previously published Neurabin/PP1 holophosphatase structure (9PDB

*Figure 6 continued on next page*

Figure 6 continued

3HVQ) (**Ragusa et al., 2010**). PP1 in white surface representation, Neurabin in ribbon representation (lilac, PP1-4E-BP1/Neurabin; red, Neurabin/PP1). 4E-BP1 in blue stick representation, unresolved sequences not shown. Structures are superimposed on PP1 residues 7–298 (rmsd = 0.21 Å, 277 alpha carbons). (**D**) Close-up view of interactions between 4E-BP1 C-terminal sequences (blue sticks) with the Neurabin PDZ domain (lilac cartoons). (**E**) AlphaFold3 model of the phosphorylated PP1-4E-BP1 chimera/Neurabin(423–593) interaction. A close-up view of predicted interaction of pT70 with the PP1 catalytic site is shown. For PAE and pLDDT plots, see **Figure 6—figure supplement 2A**. PP1 and Neurabin are shown respectively in white and lilac surface representation, with PP1 active site $Mn^{2+}$ ions in purple. 4E-BP1 sequences are in stick representation, colour-coded according to the AlphaFold3 pLDDT score (inset). See also **Figure 6—figure supplement 2C**. (**F**) AlphaFold3 modelling of the Neurabin(423–593)/PP1-5x phospho-4E-BP1 interaction. PP1 and Neurabin are shown respectively in white and lilac surface representation, with PP1 active site $Mn^{2+}$ ions in purple. 4E-BP1 sequences are in ribbon and stick representation, colour-coded according to the AlphaFold3 pLDDT score (inset), with the 4E-BP1 phosphorylations at T37, T46, S65, T70, and S101 shown in spheres. For PAE and pLDDT plots, see **Figure 6—figure supplement 2F**.

The online version of this article includes the following source data and figure supplement(s) for figure 6:

**Source data 1.** AlphaFold 3 modelling results related to **Figure 6**, **Figure 6—figure supplement 2**.

**Figure supplement 1.** Additional characterisation of the PP1-4E-BP1/Neurabin complex.

**Figure supplement 2.** AlphaFold3 predictions for phosphorylated 4E-BP1 binding to PP1 and Neurabin.

groove (**Figure 6D**, **Figure 6—figure supplement 1B**): 4E-BP1 F114, M116, and I118 side chains make hydrophobic contacts, while their main chain carbonyl and amide groups, and the C-terminal carboxylate, form an extensive hydrogen bonding network with Neurabin main chain residues L514, G515, I516, I518, and G520 (**Figure 6D**).

Apart from the C-terminal PBM, the 4E-BP1 T70 substrate sequences were largely unresolved in the crystal structure. Unlike the Phactr1 substrate chimera, no interactions were seen with the remodelled PP1-Neurabin hydrophobic groove; and no virtual enzyme-product complex and solvent phosphate were detected at the active site, perhaps reflecting the absence of stabilising hydrophobic groove interactions (**Figure 6B**). To gain insight into potential interactions at the PP1 catalytic site, we therefore used AlphaFold3 to model the PP1-4E-BP1 substrate chimera in its phosphorylated and unphosphorylated states (**Figure 6—figure supplements 2A and 1B**). While the 4E-BP1/PDZ interaction was correctly predicted regardless of phosphorylation status (**Figure 6E**, **Figure 6—figure supplement 2E**), only the phosphorylated substrate was predicted to interact directly with the PP1 active site. The 4E-BP1 substrate sequence [68]TK(pT70)PPR[73], predicted with high confidence, docks at the catalytic site, with K69 interacting with D220[PP1], pT70 interacting with the metal ions, H125[PP1] and R221[PP1], and P71 interacting with Y134[PP1], but the remodelled PP1 hydrophobic groove remains unoccupied (**Figure 6E**, **Figure 6—figure supplement 2C**). In contrast, AlphaFold3 gives only a low confidence prediction for the unphosphorylated substrate site D74 and L75 interacting with H125[PP1] and Y134[PP1] (**Figure 6—figure supplement 2D**). These results are consistent with the biochemical studies and support the notion that interaction with the remodelled hydrophobic groove plays no part in recognition of 4E-BP1 by the Neurabin/PP1 holoenzyme.

Finally, we used AlphaFold3 to predict interactions between Neurabin/PP1 and full-length 4E-BP1 phosphorylated on T37, T46, S65, T70, or S101 (**Figure 6F**, **Figure 6—figure supplement 2F**). All the predictions placed the C-terminus of 4E-BP1 in the ligand-binding groove of the PDZ domain, docking the phosphorylated T70 site in the PP1 active site in a similar manner to that seen in the substrate chimera predictions (**Figure 6F**; **Figure 6—figure supplement 2C**). Taken together with the crystallography and biochemical data, our observations support the view that the primary determinant of substrate specificity for Neurabin/PP1 is interaction with the PDZ domain adjacent to the RVxF-ΦΦ-R-W string.

## Discussion

The PP1 catalytic subunit possesses little intrinsic sequence specificity. Instead, it recognises specific substrates by forming holoenzymes in which it partners with a variety of PIPs to direct dephosphorylation to specific substrates. We previously showed that the interaction of the Phactr1 PIP with PP1 remodels the PP1 hydrophobic substrate groove in such a way as to allow specific recognition of substrate sequences C-terminal to the dephosphorylation site (**Fedoryshchak et al., 2020**). Phactr1 belongs to a group of PIPs that recognise PP1 using a common RVxF-ΦΦ-R-W string, all of which potentially remodel the hydrophobic groove; indeed, structural studies confirm that Neurabin and

**Table 1.** Crystallographic data and refinement statistics.

| | PDB 9GSU |
|---|---|
| Resolution range | 52.48–2.36 (2.42–2.36) |
| Space group | C222$_1$ |
| Unit cell a, b, c | 104.95 130.64 156.13 |
| α, β, γ | 90 90 90 |
| Total reflections | 1 207 183 (85 193) |
| Unique reflections | 44 376 (3 023) |
| Multiplicity | 27.2 (28.2) |
| Completeness (%) | 98.91 (91.66) |
| Mean I/sigma(I) | 5.29 (0.20) |
| Wilson B-factor | 49.25 |
| R-merge | 0.31 (12.11) |
| R-meas | 0.32 (12.33) |
| R-pim | 0.06 (2.30) |
| CC1/2 | 0.99 (0.28) |
| Reflections used in refinement | 44 019 (2 889) |
| Reflections used for R-free | 1 996 (131) |
| R$_{work}$ | 0.24 (0.36) |
| R$_{free}$ | 0.28 (0.32) |
| Number of non-hydrogen atoms | 6 180 |
| macromolecules | 6 156 |
| Ligands | 4 |
| Solvent | 20 |
| Protein residues | 787 |
| RMS (bonds) | 0.003 |
| RMS (angles) | 0.61 |
| Ramachandran favoured (%) | 95.74 |
| Ramachandran allowed (%) | 4.26 |
| Ramachandran outliers (%) | 0.00 |
| Clash score | 3.23 |
| Average B-factor | 57.04 |
| Macromolecules | 57.05 |
| Ligands | 68.46 |
| Solvent | 50.83 |

Spinophilin (PPP1R9A, PPP1R9B) do this differently from the Phactrs (*Ragusa et al., 2010*; *Fedoryshchak et al., 2020*). We identified potential substrates for the other RVxF-ΦΦ-R-W PIPs and elucidated mechanisms of substrate recognition, focussing on the Neurabin and Spinophilin PIPs.

Our previous study showed that a fusion protein comprising PP1 joined to Phactr1 sequences C-terminal to the RVxF motif maintains both the structure and the sequence specificity of the intact Phactr1/PP1 holoenzyme (*Fedoryshchak et al., 2020*). We used this fusion approach, in conjunction

with MS phosphoproteomics in 293 cells, to compare the specificities of the different Phactr family members and to identify potential substrates for other RVxF-ΦΦ-R-W PIPs. All four PP1-Phactr fusions exhibited virtually identical substrate specificities, which were substantially similar to that of the authentic Phactr1/PP1 holoenzyme, giving confidence that findings made with the fusions are also applicable to other RVXF-ΦΦ-R-W PIP/PP1 holoenzymes. It is likely that differential intracellular targeting and/or tissue-specific expression of the four different Phactr/PP1 holoenzymes allows further refinement of substrate specificity.

Expression of the other PP1-(RVxF-ΦΦ-R-W PIP) fusions revealed potential substrates for PNUTS and PP1-Neurabin/Spinophilin. Unlike the PP1-Phactr substrates, however, these exhibited no obvious sequence similarities in the substrate sequences adjacent to the targeted dephosphorylation remodelled hydrophobic groove. For PNUTS, we identified only the SET1A/B-COMPASS complex components CXXC1/CFP1 and SETD1B (*Cenik and Shilatifard, 2021*). These may be recruited to PNUTS by interaction with its binding partner WDR82, which is also present in SET1A/B-COMPASS complexes (*Cenik and Shilatifard, 2021*), but further work will be necessary to confirm this. We did not recover potential substrates for PP1-PPP1R15A/B. It is possible that the use of the PP1 fusion approach for these PIPs is confounded by interaction of these PIPs with G-actin (*Chen et al., 2015*; *Yan et al., 2021*).

Expression of the PP1-Neurabin and PP1-Spinophilin fusions identified nine potential substrates with eighteen candidate dephosphorylation sites, of which eight represented conserved mTORC1-dependent phosphorylations of the translational regulators 4E-BP1 and 4E-BP2. Using immunoblotting, we found that the p70 S6K activating phosphorylation at T389 also appears to be a substrate for PP1-Neurabin. The 4E-BPs and p70 S6K are critical targets of the mTORC1 pathway, which links translational control to cell nutrient status and extracellular signals in multiple settings (*Hoeffer and Klann, 2010*; *Liu and Sabatini, 2020*). Consistent with this, we found that overexpression of PP1-Neurabin can suppress translation when expressed in 293 cells. It is therefore likely that Neurabin and Spinophilin can negatively regulate both arms of the mTORC1 protein synthesis pathway (*Figure 3—figure supplement 1C*).

The presence of multiple dephosphorylation sites on the 4E-BPs with no obvious primary sequence homology raised the question of how these targets are recognised. We found that dephosphorylation of 4E-BP1 by PP1-Neurabin requires its C-terminal sequences, which constitute a classical PBM (*Harris and Lim, 2001*; *Tonikian et al., 2008*; *Subbaiah et al., 2011*). The micromolar affinity of this interaction, which is stronger than that previously reported for p70 S6K (*Burnett et al., 1998*), may reflect the necessity for effective competition with the mTORC1 kinase complex, which binds the 4E-BPs using a C-terminal TOS motif that overlaps the PBM (*Schalm and Blenis, 2002*; *Yang et al., 2017*). Although we recovered a number of other potential Neurabin and Spinophilin substrates in the proteomics screen, none contains an obvious potential C-terminal PBM; further work will be necessary to establish whether they are bona fide substrates. We found that other reported ligands, such as the glutamatergic and dopaminergic receptors (*Kelker et al., 2007*), bind the Neurabin/Spinophilin PDZ domains with much lower affinity than 4E-BP1 or even p70 S6K. In neurons, the association of these receptors with Neurabin and PP1 correlates with channel activity and dephosphorylation (*Wu et al., 2008*; *Yan et al., 1999*, reviewed by *Foley et al., 2021*). It may be that Neurabin/Spinophilin oligomerisation and/or F-actin binding, and membrane localisation, effectively increases the avidity of such weak PBM-PDZ interactions, but more work is required to establish whether these channels are direct Neurabin/PP1 substrates.

We used synthetic peptide substrates derived from 4E-BP1, or the Phactr1/PP1 substrate IRSp53, to investigate how the substrate recognition mechanisms differ between the PP1-Neurabin and PP1-Phactr fusions. In the context of the PP1-PIP fusions, and by extension in that of the PIP/PP1 holoenzyme, the PIP sequences play a critical role in directing substrate specificity and potentiating catalytic efficiency. In the PP1-Neurabin fusion, interaction with the PDZ domain is the critical determinant of both catalytic efficiency and specificity (*Figure 5—figure supplement 2*). It increases catalytic efficiency 100-fold above that seen with PP1 alone or the PP1-Phactr1 fusion. Moreover, the PP1-Neurabin fusion was inactive with the Phactr1/PP1 substrate IRSp53 unless the latter was made competent to bind the Neurabin PDZ domain. Our observations suggest a model in which 4E-BPs are recruited to Neurabin/PP1 via high-affinity PBM-PDZ interaction, with the relatively low affinity of active site interaction allowing dephosphorylation of the multiple sites during one round of PDZ

binding (*Figure 5—figure supplement 2*). In contrast, dephosphorylation of the Phactr1/PP1 target IRSp53 pS455 by the PP1-Phactr1 fusion protein was critically dependent on interaction with the remodelled hydrophobic groove, consistent with our previous data (*Figure 5—figure supplement 2*; *Fedoryshchak et al., 2020*).

Several lines of evidence support the view that interaction with the remodelled PP1 hydrophobic groove plays no role in PP1-Neurabin substrate recognition. First, the PP1-Neurabin substrates we identified have no obvious sequence similarity at positions +3 to +6, the sequences that potentially interact with the remodelled hydrophobic groove, and alanine substitutions at these positions do not influence PP1-Neurabin catalytic activity. Structural analysis of a PP1-4E-BP1 substrate fusion complexed with Neurabin also revealed no substrate contacts with PP1, although the PDZ-4E-BP1 interaction was well resolved. Finally, AlphaFold modelling of complexes formed between 4E-BP1 substrates and the Spinophilin/PP1 holoenzyme predicted phosphorylation-specific interactions with the PP1 catalytic site, but no interactions with the remodelled hydrophobic groove. We were intrigued to note, however, that alanine substitutions at +1 and +2 in relative to 4E-BP1 pT70 increased reactivity with both PP1-Neurabin and PP1-Phactr1. We speculate that this 'suboptimal' reactivity may reflect a requirement to balance dephosphorylation rates between the multiple 4E-BP1 phosphorylation sites, especially if multiple rounds of dephosphorylation occur for each PBM-PDZ interaction.

Although the remodelled hydrophobic groove does not play a role in 4E-BP recognition by Neurabin/PP1, this does not necessarily mean that sequence-specific dephosphorylation by other PIP/PP1 complexes and substrate recruitment by protein interactions are inherently mutually exclusive: in principle, a PIP could recruit substrates through protein interactions and remodel the PP1 substrate binding grooves to allow discrimination between different phosphorylated sites on the target protein. That said, we think it likely that the other RVxF-ΦΦ-R-W PIPs will also recruit substrates by interaction with other PIP domains or proteins bound to them. Such interactions might involve WDR82 for PNUTS and G-actin for PPP1R15A/B (*Yan et al., 2021*; *Erickson et al., 2024*).

Our findings establish Neurabin/PP1 and Spinophilin/PP1 holoenzymes as new candidate regulators of the mTORC1 pathway. Many studies have focussed on mTORC1 regulation of translation on the control of cell growth, particularly in cancer settings (*Liu and Sabatini, 2020*), and previous work has shown that endogenous PP2Cγ/ PPM1G contributes to 4E-BP1 dephosphorylation in 293 and HCT116 cells (*Liu et al., 2013*). Localised translational regulation is also important in other settings, however, including neuronal development and function (*Holt et al., 2019*). Here, mTORC1 signalling plays an important role in neuronal plasticity (*Hoeffer and Klann, 2010*), as do both Neurabin and Spinophilin, which are largely neuron-specific and enriched in dendritic spines (*Wu et al., 2008*, reviewed by *Foley et al., 2021*), and the 4E-BPs, particularly 4E-BP2 (for references, see *Aguilar-Valles et al., 2015*). Our finding that 4E-BPs are Neurabin/PP1 substrates potentially establishes a direct link between Neurabin and neuronal mTORC1 signalling (*Figure 3—figure supplement 1C*), which will be an interesting topic for future investigation.

## Methods
### Plasmids

NEBuilder HiFi DNA Assembly Cloning Kit and NEB Q5 Site-Directed Mutagenesis Kit were used according to the manufacturer's protocols for plasmid assembly and mutagenesis. All primers are listed in *Supplementary file 1*.

PIP sequences were amplified from 293 cells cDNA, except for the Neurabin sequence which was commercially synthesised. pET28-PP1(7–300) and pET28-PP1(7–304)-SGSGS-Phactr1(526–580) Phactr1 plasmids were as described (*Fedoryshchak et al., 2020*). Other fusion proteins were expressed. pET28-based expression plasmids were used to express other PP1-PIP fusions as follows: PP1(7–304)-SGSGS-Phactr2(580–634); PP1(7–304)-SGSGS-Phactr3(505–559); PP1(7–304)-SGSGS-Phactr4(648-702); PP1(7–304)-SGSGS-Neurabin(464–610); PP1(7–304)-SGSGS-Spinophilin(456–602); PP1(7–304)-SGSGS-R15A(562–674); PP1(7–304)-SGSGS-R15B(647–713); PP1(7–304)-SGSGS-PNUTS(408–619).

For T-REx cell lines, pOG44 (Thermo) and pcDNA5/FRT/TO plasmids (Thermo) were used in conjunction with N-terminally Flag-tagged PP1-PIP fusions which were inserted into pcDNA5/FRT/TO.

pcDNA3.1 IRSp53 and pcDNA3.1 IRSp53 L460A were as described (*Fedoryshchak et al., 2020*). To obtain pEF mCherry-4E-BP1, the 4E-BP1 sequences were amplified from 293 cells and inserted into

the pEF-mCherry plasmid. Site-directed mutagenesis was used to derive mutants ΔPDZ, 118+A, S65A, S101A, S65A/S101A. pGEX-Neurabin(PDZ) and pGEX-Spinophilin(PDZ) plasmids were obtained by cloning Neurabin and Spinophilin PDZ domains into the pGEX 6P2 vector (GE Healthcare). The PP1-4E-BP1 chimera was generated by insertion of 4E-BP1 sequences into pET28-PP1.

## Cell lines and transfections

Commercially available 293 Flp-In T-REx cells and pOG44+pcDNA5/FRT/TO stably transfected derivatives were used throughout. Cells were maintained in a humidified incubator at 37°C and 5% CO$_2$ and cultured in DMEM (Gibco) supplemented with 10% FCS (Gibco) and penicillin-streptomycin (Sigma). Prior to stable transfection, 293 Flp-In T-REx cells were maintained with 100 mg/ml Zeocin (Invivogen). Stably transfected 293 Flp-In T-REx derivative cell lines were cultured in a medium supplemented with 5 mg/ml Blasticidin (Invivogen) and 100 mg/ml Hygromycin B (Invivogen). The 293 Flp-In T-REx cell line was obtained through the Francis Crick Institute Cell Services repository. The cell line was authenticated and regularly tested for mycoplasma contamination (negative).

For stable transfection of a PP1-fusion protein, pOG44 and pcDNA5/FRT/TO-PP1-fusion plasmids were mixed at a ratio of 9:1. Lipofectamine 2000 (Invitrogen) in Opti-MEM (Gibco) was added, and transfection was done following the manufacturer's protocol. 100 μg/ml Hygromycin B (Invitrogen) was added to start the selection of stable cell line 2 days after transfection. Selection was complete, and cell line stocks were frozen after 14 days.

pEF IRSp53 and pEF-mCherry-4E-BP1 plasmids were transfected with Lipofectamine 2000 (Invitrogen) in Opti-MEM (Gibco) following the manufacturer's protocol. Cells were lysed 1 day after transfection in a buffer containing 20 mM Tris pH 7.4, 150 mM NaCl, 1 mM EDTA, 1 mM EGTA, 1% Triton X-100, 10% glycerol, 0.2% SDS. Lysates were cleared by centrifugation. 4× LDS sample buffer supplemented with DTT was added before running immunoblots.

Unless otherwise indicated, 1000 nM of tetracycline was used to express the PP1 fusion proteins in stably transfected 293 Flp-In T-REx cells, and the experiments were performed 16 hr post-induction.

## Immunoblotting

SDS-PAGE analysis of cell lysates and immunoblotting was performed using standard techniques; the signal was visualised and quantified using Odyssey CLx instrument (LI-COR) and the Image Studio (LI-COR) Odyssey Analysis Software. Primary antibodies used were Flag (1:2000, clone M2, Sigma F7425, mouse), IRSp53 (1:1000, Abcam ab15697), IRSp53 pS455 (1:500, previously described in *Fedoryshchak et al., 2020*), Afadin (1:200, Santa Cruz sc-74433), Afadin pS1275 (1:500, previously described in *Fedoryshchak et al., 2020*), GAPDH (1:2000, clone 0411, Santa Cruz sc-47724), 4E-BP1 (1:1000, Cell Signaling 9452), 4E-BP1 pT37/46 (1:1000, Cell Signaling 9459), 4E-BP1 pS65 (1:1000, Cell Signaling 9451), 4E-BP1 pT70 (1:1000, Cell Signaling 9455), mCherry (1:1000, clone 16D7, Thermo M11240, rat), S6K (1:1000, Cell Signaling 9202), S6K pS371 (1:1000, Cell Signaling 9208), S6K pT389 (1:1000, Cell Signaling 9205), S6K pT421/pS424 (1:1000, Cell Signaling 9204). Secondary antibodies labelled with IRDye 800CW and IRDye 680LT were from LI-COR.

## Proteomics

Total and phospho-proteomics experiment was performed according to a detailed protocol previously published (*Jones et al., 2020*). Cells expressing PP1-PIP fusion proteins, PP1 only, or vector alone were induced with tetracycline for 16 hr. Cells were lysed in buffer containing 8 M urea, 50 mM HEPES pH 8.5, 10 mM glycerol 2-phosphate, 50 mM NaF, 5 mM sodium pyrophosphate, 1 mM EDTA, 1 mM sodium vanadate, 1 mM dithiothreitol, 1:50 protease inhibitor cocktail (Roche), 1:100 phosphatase inhibitor cocktail, 400 nM okadaic acid. Cysteines were reduced and alkylated by iodoacetamide followed by trypsin/rLysC protease digestion. Peptide samples were labelled with 10-plex (UK288606) and additionally 131C (VC294053) TMT reagents from Thermo and pooled. Part of the sample was injected and saved for the total proteome analysis. The rest of the sample was used for two-step phosphopeptide enrichment on TiO$_2$ beads (Thermo) and FeNTA beads (Thermo). Phosphopeptides were fractionated. Total proteome and enriched phosphopeptide fractions were separated on a 50 cm, 75 μm I.D. Pepmap column over a 2 hr gradient and eluted directly into the Orbitrap Fusion Lumos, operated with Xcalibur software, with measurement in MS2 and MS3 modes. The instrument

was set up in data-dependent acquisition mode, with the top 10 most abundant peptides selected for MS/MS by HCD fragmentation.

Raw mass spectrometric data were processed in MaxQuant (v1.6.12.0); database search against the *Homo sapiens* canonical sequences from UniProtKB was performed using the Andromeda search engine. Fixed modifications were set as carbamidomethyl (C) and variable modifications set as oxidation (M), acetyl (protein N-term), and phospho (STY). The estimated false discovery rate was set to 1% at the peptide, protein, and site levels, with a maximum of two missed cleavages allowed. Reporter ion MS2 or reporter ion MS3 was appropriately selected for each raw file.

Phosphorylation site tables were imported into Perseus (v1.6.14.0) for analysis. Contaminants and reverse peptides were cleaned up from the phosphosites (STY), and the values normalised and averaged between MS2 and MS3 datasets using the Z-score function across columns. All samples were analysed as triplicates, except PP1-Spinophilin and PP1-PNUTS, for which duplicates were used due to the presence of an outlier dataset as determined by principal component analysis. Phosphorylation site enrichments were compared using multiple t-tests with permutation-based false discovery cut-off at 5% unless indicated otherwise. Enrichment data are summarised in *Supplementary file 2*.

Mass spectrometry proteomics data have been deposited to the ProteomeXchange Consortium via the PRIDE partner repository with the dataset identifier PXD055166.

## Peptides

Peptides were synthesised by the Francis Crick Institute Chemical Biology Science Technology Platform using standard Fmoc-SPPS techniques.

| Name | Sequence (FAM, 6-carboxyfluorescein; eahx, aminohexanoic acid linker; OH, carboxyl at C-terminus) |
|---|---|
| 4EBP | FAM-eahx-QFEMDI-OH |
| 4EBP-mut | FAM-eahx-QFESGS-OH |
| S6K | FAM-eahx-EHLRMNL-OH |
| Kalirin-7 | FAM-eahx-DPFSTYV-OH |
| GluR1 | FAM-eahx-LGATGL-OH |
| GluR2 | FAM-eahx-IESVKI-OH |
| GluN1-1 | FAM-eahx-SVSTVV-OH |
| GluN2A | FAM-eahx-SIESDV-OH |
| 4E-BP1[PBM] | Ac-NSPVTK(**pT**)PPRDLPTIGSGSQFESGS-**OH** |
| 4E-BP1[LLD-PBM] | Ac-NSPVTK(**pT**)PPR**LLD**TIGSGSQFEMDI-**OH** |
| IRSp53[PBM] | Ac-QQGKSS(**pS**)TGNLLDKDGSGSQFEMDI-**OH** |
| IRSp53[MUT] | Ac-QQGKSS(**pS**)TGNLLDKDGSGSQFESGS-**OH** |
| IRSp53[5A-PBM] | Ac-QQGKSS(**pS**)TGNL**A**DKDGSGSQFEMDI-**OH** |
| IRSp53[5A-MUT] | Ac-QQGKSS(**pS**)TGNL**A**DKDGSGSQFESGS-**OH** |
| 4E-BP1[12A-PBM] | Ac-NSPVTK(**pT**)**AA**RDLPTIGSGSQFEMDI-**OH** |
| 4E-BP1[34A-PBM] | Ac-NSPVTK(**pT**)PP**AA**LPTIGSGSQFEMDI-**OH** |
| 4E-BP1[56A-PBM] | Ac-NSPVTK(**pT**)PPRD**AA**TIGSGSQFEMDI-**OH** |
| 4E-BP1[1A-PBM] | Ac-NSPVTK(**pT**)**A**PRDLPTIGSGSQFEMDI-**OH** |
| 4E-BP1[2A-PBM] | Ac-NSPVTK(**pT**)P**A**RDLPTIGSGSQFEMDI-**OH** |

## Protein expression and purification

PP1-fusion proteins were produced as 6xHis-tagged fusion proteins in (DE3) *Escherichia coli* cells (Invitrogen) with pGRO7 co-expression as described (*Choy et al., 2014*). Overnight pre-cultures (400 ml) were grown in LB medium supplemented with 1 mM MnCl$_2$ and used to inoculate a 100 l fermenter. After growth to OD$_{600}$ of ~0.5, 2 g/l of arabinose was added to induce GroEL/GroES expression. At

OD$_{600}$ ~1, the temperature was lowered to 17°C and protein expression induced with 0.1 mM IPTG for ~18 hr. Cells were harvested, re-suspended in fresh LB medium/1 mM MnCl$_2$/200 µg/ml chloramphenicol, and agitated for 2 hr at 17°C. Harvested cells were resuspended in lysis buffer (50 mM Tris-HCl, pH 8.5, 5 mM imidazole, 700 mM NaCl, 1 mM MnCl$_2$, 0.1% vol/vol TX-100, 0.5 mM TCEP, 0.5 mM AEBSF, 15 µg/ml benzamidine, and complete EDTA-free protease inhibitor tablets), lysed by French Press, clarified, and stored at –80°C.

Clarified lysates were loaded onto a 5 ml HisTrap crude column on an AktaPure FPLC, washed with 20CV of buffer A (25 mM Tris-HCl pH 8.5, 250 mM NaCl, 10 mM imidazole pH 8, 1 mM MnCl$_2$), His-tagged fusion proteins were eluted in buffer B (buffer A+240 mM imidazole) and purified using size exclusion chromatography on a Superdex 200 26/60 column in SEC buffer1 (25 mM Tris-HCl pH 8.5, 200 mM NaCl, 0.5 mM TCEP, 10 mM imidazole). The His tag was then cleaved off by incubating overnight with His-Tev protease at 4°C, and the cleaved product recovered by passage of the sample over a 5 ml HisTrap crude column. Protein was then concentrated and further purified on a Superdex 75 equilibrated in SEC buffer2 (25 mM Tris-HCl pH 8.5, 200 mM NaCl, 0.5 mM TCEP). The PP1-fusion proteins were concentrated to 10 mg/ml and stored at –80°C.

His-tagged Neurabin and Spinophilin PDZ domains were produced in BL21 (DE3) *E. coli* cells (Invitrogen). Overnight pre-cultures were grown in LB medium, 15 ml was used to inoculate 1 l of TB media in 2 l baffled flasks. When OD$_{600}$=1, the temperature was lowered to 20°C, and protein expression induced by addition of 0.1 mM IPTG for ~18 hr. Harvested cells were resuspended in lysis buffer (50 mM Tris-HCl, pH 8.5, 5 mM imidazole, 700 mM NaCl, 0.1% vol/vol TX-100, 0.5 mM TCEP, 0.5 mM AEBSF, 15 µg/ml benzamidine, and complete EDTA-free protease inhibitor tablets), lysed by French press, clarified, and stored at –80°C. PDZ domains were purified using the same protocol as PP1-fusion proteins without addition of 1 mM MnCl$_2$ in the buffers.

## Crystallisation and structure determination

PP1-4E-BP1/Neurabin was concentrated to 10 mg/ml and crystallised at 20°C using sitting-drop vapour diffusion. Sitting drops of 1 µl consisted of a 1:1 (vol:vol) mixture of protein and well solution (20% PEG 6000, 0.2 M MgCl$_2$, 0.1 M MES pH 6.0). Crystals appeared within 5 days and reached maximum size after 7 days. Crystals were cryoprotected in well solution supplemented with 15% glycerol+15% ethylene glycol and flash-frozen in liquid nitrogen. 100 K at beamlines I04 (mx25587-44) of the Diamond Light Source Synchrotron (Oxford, UK). Data collection and refinement statistics are summarised in *Table 1*. Datasets were indexed, scaled, and merged with xia2 (*Winter et al., 2013*). Molecular replacement used the atomic coordinates of human PP1 from PDB 4M0V (*Choy et al., 2014*) in PHASER (*McCoy et al., 2007*). Refinement used Phenix (*Adams et al., 2010*). Model building used COOT (*Emsley et al., 2010*) with validation by PROCHECK (*Vaguine et al., 1999*). Two copies of the complex were modelled in the asymmetric unit, but the entire PDZ domain of one copy was poorly defined in the density and therefore not modelled. The same issue was previously observed for the unliganded Spinophilin/PP1 structure (*Ragusa et al., 2010*). For structure analysis, we used the second copy of the complex showing well-resolved density for PP1-4E-BP1 chimera and Neurabin. AlphaFold3 predictions (*Abramson et al., 2024*) were performed using the AlphaFold3 server. Output structure prediction and parameter files are presented in *Figure 6—source data 1*.

## Fluorescence polarisation

FAM-labelled peptides were dissolved in a buffer 25 mM Tris-HCl pH 8, 250 mM NaCl, 0.5 mM TCEP. Peptide concentration was measured using Thermo Scientific NanoDrop One by FAM fluorescence at 495 nm. FP assays (10 µl final volume) were performed in 384-well plates. 2 µl of 500 nM peptide solutions were added to each well (100 nM final concentration). 8 µl of Neurabin PDZ or Spinophilin PDZ was added as a serial dilution, starting at 1000 µM (800 µM final concentration). Anisotropies were read out on BMG Labtech CLARIOstar Plus microplate reader. Binding constants were estimated in GraphPad Prism 8 by fitting readouts with the following equation:

$$A = A_f + \left(A_b - A_f\right) * \left(K_d + L + C - \frac{\left(K_d + L + C\right)^{0.5}}{2 * L}\right)$$

(A, anisotropy measured; $A_f$, anisotropy of free peptide; $A_b$, anisotropy of bound peptide; L, labelled peptide concentration; C, protein concentration (X axis); $K_d$, binding constant).

## Phosphatase activity assays

Phosphopeptides were dissolved in 25 mM Tris-HCl pH 8, 250 mM NaCl buffer. Peptide concentration was measured using DeNovix DS-11 based on absorbance values at 215 nm. Assays (10 μl final volume) were performed in 384-well plates. The activity of the PP1 fusion preparation was established on the day of each experiment, using 50 μM pIRSp53$^{PBM}$ peptide as standard, with PP1-Phactr1 and PP1-Neurabin at various dilutions.

Peptides (4 μl) were serially diluted twofold from 1 mM in complex buffer (1 mM $MnCl_2$, 25 mM Tris-HCl pH 8, 250 mM NaCl, 0.5 mM TCEP). 2 μl of 5x Phosphate Sensor (Thermo) was added to the wells. 2 μl of PP1-fusion protein was added to the wells, and fluorescence measurements were started immediately and taken every 3 min or 5 min using BMG Labtech CLARIOstar Plus microplate reader. Data was collected for 15 min at room temperature. Phosphate standards were measured to convert fluorescence readouts into phosphate concentration using a standard curve. Difference in fluorescence between the last and first data point was used as a readout.

A concentration of PP1-Phactr1 and PP1-Neurabin was chosen such that readouts were in the linear range of the phosphate sensor detection. To measure $K_M$ and catalytic efficiency, an assay with varying dilutions of phosphopeptides was set up. 4 μl of a corresponding peptide dilution was added to wells, followed by 2 μl of 5x Phosphate Sensor. 2 μl of PP1-fusion protein was added to the wells, and fluorescence measurements were started immediately and taken every 3 min (alternatively, every 5 min). Data was collected for 15 min at room temperature. Phosphate standards were measured to convert fluorescence readouts into phosphate concentration using a standard curve. Difference in fluorescence between the last and first data point was used as a readout. Rate constants were estimated in GraphPad Prism 8 by fitting readouts to the rearranged Michaelis-Menten equation:

$$\frac{P}{t * E} = \left[\frac{k_{cat}}{k_M}\right] * \frac{C}{\left(\frac{C}{k_M} + 1\right)}$$

(P, phosphate released per time t; E, PP1-fusion concentration; $[k_{cat}/K_M]$, catalytic efficiency; C, initial phosphopeptide concentration; $K_M$, Michaelis constant). Significance comparisons between activities with different peptides, or between different enzymes for the same substrate peptide, were calculated using a two-tailed two-sample equal variance Student's t-test. The assay data are presented in *Supplementary file 3*.

## 4E-BP1 dephosphorylation immunoblot assay

293 Flp-In T-REx cells were transfected with mCherry-4E-BP1 (WT or 118+A mutant). Cells were lysed on the following day in the Tris-Triton buffer (see above). mCherry-tagged constructs were enriched from the lysates using RFP-trap magnetic agarose beads (ChromoTek, rtma-20) using the manufacturer's protocol. Enriched fractions were eluted from the beads with 100 μl 200 mM glycine pH 2.5, neutralised with 20 μl 1M Tris pH 10.4, and 400 μl of 25 mM Tris-HCl pH 8, 250 mM NaCl buffer added resulting in pure phosphorylated solution of mCherry-4E-BP1 (WT or 118+A mutant). 8 μl of recombinantly expressed PP1-Neurabin (dilutions to provide 0-1000 nM final concentration) were added to 40 μl aliquots of mCherry-4E-BP1 (WT or 118+A mutant), and incubated for 15 min at room temperature before addition of 18 μl 4x LDS sample buffer with DTT. The samples were warmed to 70°C for 10 min before analysis by immunoblotting.

## Protein synthesis assay

293 Flp-In T-REx cells were cultured in a six-well plate. PP1, PP1-Neurabin, or vector construct expression were induced 16 hr before the experiment with 1 μM tetracycline. Rapamycin was used at 50 nM for 16 hr. *O*-propargyl puromycin (OPP; 5 μM) was added to the culture for 30 min. Then, cells were washed once with PBS and resuspended using trypsin. Cells were collected by centrifugation at 300×*g* for 5 min, then washed once with PBS and 4% PFA was added for 15 min. Cells were collected and resuspended in 0.5% Triton X-100 in PBS for 15 min. Cells were collected and resuspended in 2% BSA in PBS twice. Click-reaction mixture was prepared as follows: 1540 μl of water, 200 μl of 10× PBS, 40 μl

CuSO$_4$ (20 mM) and BTTAA (100 mM) mixture, 20 µl of 5 mM Alexa Fluor-488 azide, 200 µl 100 mM sodium ascorbate.

200 µl of click-reaction mixture was added to each sample and incubated for 30 min in the dark. Cells were collected and resuspended in 10 mM EDTA. Then, cells were incubated for 30 min in H33342 1:10,000 in PBS. Cells were washed with PBS and analysed on BD LSRFortessa Cell Analyzer. FlowJo software was used to analyse flow cytometry data. Single-cell readouts were isolated using forward and side scatter parameters. Mean fluorescent intensity in the Alexa Fluor-488 channel was used as an assay readout.

## Acknowledgements

We thank Sila Ultanir for helpful discussions, insights into neuronal plasticity, and support for pilot experiments in neurons, and lab members, Neil McDonald, and Sila Ultanir for helpful discussions and comments on the manuscript. We thank Helen Flynn from Crick Proteomics STP for assistance with phospho-proteomics experiments, Jo Redmond from Chemical Biology STP for help with discussions about substrate specificity and assistance with peptide synthesis, and Simone Kunzelman from the Structural Biology STP for assistance with FP assay. This work was supported by the Francis Crick Institute which receives its core funding from Cancer Research UK (CC2102), the UK Medical Research Council (CC2102), and the Wellcome Trust (CC2102). This research was funded in whole, or in part, by the Wellcome Trust CC2102. For the purpose of Open Access, the author has applied a CC BY public copyright licence to any Author Accepted Manuscript version arising from this submission. The authors have no conflicts of interest.

## Additional information

### Funding

| Funder | Grant reference number | Author |
|---|---|---|
| Cancer Research UK | CC2102 | Richard Treisman |
| Medical Research Council | CC2102 | Richard Treisman |
| Wellcome Trust | CC2102 | Richard Treisman |

The funders had no role in study design, data collection and interpretation, or the decision to submit the work for publication. For the purpose of Open Access, the authors have applied a CC BY public copyright license to any Author Accepted Manuscript version arising from this submission.

### Author contributions

Roman O Fedoryshchak, Conceptualization, Formal analysis, Investigation, Visualization, Methodology, Writing - original draft, Writing - review and editing; Karim El-Bouri, Resources, Investigation; Dhira Joshi, Resources, Methodology; Stephane Mouilleron, Supervision, Investigation, Visualization, Methodology, Writing - review and editing; Richard Treisman, Conceptualization, Supervision, Funding acquisition, Visualization, Methodology, Writing - original draft, Writing - review and editing

### Author ORCIDs

Roman O Fedoryshchak ⓘ https://orcid.org/0000-0003-1865-8372
Karim El-Bouri ⓘ https://orcid.org/0000-0002-4542-0856
Dhira Joshi ⓘ https://orcid.org/0000-0001-8660-2528
Stephane Mouilleron ⓘ https://orcid.org/0000-0001-7977-6298
Richard Treisman ⓘ https://orcid.org/0000-0002-9658-0067

Reviewer #1 (Public review): https://doi.org/10.7554/eLife.103403.3.sa1
Reviewer #2 (Public review): https://doi.org/10.7554/eLife.103403.3.sa2
Reviewer #3 (Public review): https://doi.org/10.7554/eLife.103403.3.sa3
Author response https://doi.org/10.7554/eLife.103403.3.sa4

## Additional files

### Supplementary files

Supplementary file 1. Primer sequences. (**A**) Primers for cloning new phosphatase fusions into pET28 vector. (**B**) Subcloning phosphatase fusion into pcDNA5. (**C**) Primers for cloning 4E-BP1 constructs into pEF vector. (**D**) Primers for 4E-BP1 mutagenesis. (**E**) Primers for subcloning PDZ domain constructs into pGEX vector. (**F**) Primers for the PP1-4E-BP1 chimera assembly.

Supplementary file 2. Proteomics. (**A**) Phosphoproteomics raw intensities based on MaxQuant analysis. (**B**) Phosphoproteomics processed, cleaned-up and normalised data used for further analyses. (**C**) Quantification of phosphorylation site preferences for PP1 (PP1 vs empty). (**D**) Substrate preferences of Phactr1-4-PP1 fusions as opposed to PP1 (Phactrs vs PP1). (**E**) Identification of substrates for Neurabin, Spinophilin, PNUTS, R15A and R15B PP1 fusions. (**F**) Total proteomics raw data and enrichment of proteins in the PP1-Neurabin samples.

Supplementary file 3. Activity assay data. (**A**) Comparative overview of significant differences between reactions performed with different peptides, or between different enzymes for the same substrate peptide. (**B**) Combined catalytic efficiency data from all 6 replicate assays. Raw and normalised values are presented separately. Normalization was performed by normalisation of average catalytic efficiency, using PBM peptides for PP1-Neurabin and IRSp53 peptides for PP1-Phactr1. (**C**) Combined Michaelis-Menten constants $K_M$ data from all 6 replicate assays. (**D-H**) Curve-fitting parameters for each replicate assay.

MDAR checklist

Source data 1. All plasmid sequences used.

### Data availability

Mass spectrometry proteomics data have been deposited to the ProteomeXchange Consortium via the PRIDE partner repository with the dataset identifier PXD055166. Atomic coordinates and crystallographic structure factors for PP1-Neurabin bound to 4E-BP1 have been deposited in the Protein Data Bank under the accession code PDB 9GSU. All other data generated or analysed during this study are included in the manuscript and supporting files.

The following datasets were generated:

| Author(s) | Year | Dataset title | Dataset URL | Database and Identifier |
|---|---|---|---|---|
| Mouilleron S, Treisman R, Fedoryshchak R, Elbouri K | 2025 | Structure of PP1-Neurabin bound to 4E-BP1 | https://doi.org/10.2210/pdb9gsu/pdb | Worldwide Protein Data Bank, 10.2210/pdb9gsu/pdb |
| Fedoryshchak R, Treisman R | 2025 | Identification of PP1-fusion phosphatase substrates | https://www.ebi.ac.uk/pride/archive/projects/PXD055166 | PRIDE, PXD055166 |

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

# Appendix 1

**Appendix 1—key resources table**

| Reagent type (species) or resource | Designation | Source or reference | Identifiers | Additional information |
|---|---|---|---|---|
| Gene (*Homo sapiens*) | PPP1CA | UniProt P62136 | | |
| Gene (*H. sapiens*) | Phactr1 | UniProt Q9C0D0 | | |
| Gene (*H. sapiens*) | Phactr2 | UniProt O75167 | | |
| Gene (*H. sapiens*) | Phactr3 | UniProt Q96KR7 | | |
| Gene (*H. sapiens*) | Phactr4 | UniProt Q8IZ21 | | |
| Gene (*H. sapiens*) | PPP1R10 (PNUTS) | UniProt Q96QC0 | | |
| Gene (*H. sapiens*) | PPP1R9A Neurabin | UniProt Q9ULJ8 | | |
| Gene (*H. sapiens*) | PPP1R9B Spinophilin | UniProt Q96SB3 | | |
| Gene (*H. sapiens*) | PPP1R15A | UniProt O75807 | | |
| Gene (*H. sapiens*) | PPP1R15B | UniProt Q5SWA1 | | |
| Gene (*H. sapiens*) | S6K (RPS6KB1) | UniProt P23443 | | |
| Gene (*H. sapiens*) | 4E-BP1 (EIF4EBP1) | UniProt Q13541 | | |
| Strain (*Escherichia coli*) | 5-alpha competent *E. coli* | NEB C2992I | | |
| Strain (*E. coli*) | Protein expression BL21 (DE3) | NEB C2527H | | |
| Cell line (*H. sapiens*) | 293 Flp-In T-REx | Thermo R78007 | RRID:CVCL_U427 | |
| Antibody | Rabbit polyclonal anti-Flag M2 | Sigma F7425 | RRID:AB_439685 | WB 1:500 |
| Antibody | Goat polyclonal anti-IRSp53 | Abcam ab15697 | RRID:AB_301929 | WB 1:500 |
| Antibody | Mouse monoclonal anti-Afadin (B-5) | Santa Cruz sc-74433 | RRID:AB_1118816 | WB 1:200 |
| Antibody | Mouse monoclonal anti-GAPDH (0411) | Santa-Cruz sc-47724 | RRID:AB_627679 | WB 1:2000 |
| Antibody | Rabbit polyclonal anti-IRSp53 pS455 | *Fedoryshchak et al., 2020* | | WB 1:500 |
| Antibody | Rabbit polyclonal anti-afadin pS1282 | *Fedoryshchak et al., 2020* | | WB 1:500 |
| Antibody | Rabbit polyclonal anti-4E-BP1 pT70 | Cell Signaling 9455 | RRID:AB_330947 | WB 1:1000 |
| Antibody | Rabbit polyclonal anti-4E-BP1 pS65 (pS101) | Cell Signaling 9451 | RRID:AB_330945 | WB 1:1000 |
| Antibody | Rabbit polyclonal anti-4E-BP1 pT37/46 | Cell Signaling 9459 | RRID:AB_330944 | WB 1:1000 |
| Antibody | Rabbit polyclonal anti-4E-BP1 | Cell Signaling 9452 | RRID:AB_330946 | WB 1:1000 |
| Antibody | Rat monoclonal anti-mCherry | Thermo M11240 | RRID:AB_2536611 | WB 1:1000 |
| Antibody | Rabbit polyclonal anti-S6K pT389 | Cell Signaling 9205 | RRID:AB_330944 | WB 1:1000 |
| Antibody | Rabbit polyclonal anti-S6K pS371 | Cell Signaling 9208 | RRID:AB_331680 | WB 1:1000 |
| Antibody | Rabbit polyclonal anti-S6K pT421/pS424 | Cell Signaling 9204 | RRID:AB_331679 | WB 1:1000 |
| Antibody | Rabbit polyclonal anti-S6K | Cell Signaling 9202 | RRID:AB_331676 | WB 1:1000 |
| Antibody | IRDye 680RD Secondary Antibodies | Licor 925-68073 | RRID:AB_2716687 | WB 1:20,000 |
| Antibody | IRDye 800CW Secondary Antibodies | Licor 925-32214 | RRID:AB_2814909 | WB 1:20,000 |
| Recombinant DNA reagent (plasmid) | pcDNA3.1 IRSp53 | Dr. Eunjoon Kim; PMID: 15673667 | | |
| Recombinant DNA reagent (plasmid) | pcDNA3.1 IRSp53 L460A | *Fedoryshchak et al., 2020* | | See Methods; *Figure 5B* |

*Appendix 1 Continued on next page*

*Appendix 1 Continued*

| Reagent type (species) or resource | Designation | Source or reference | Identifiers | Additional information |
|---|---|---|---|---|
| Recombinant DNA reagent (plasmid) | pTRIPZ | *Esnault et al., 2014* | | |
| Recombinant DNA reagent (plasmid) | pGEX 6P2 | GE Healthcare 27-4598-01 | | |
| Recombinant DNA reagent (plasmid) | pET28 PP1(7–300) | Dr. Wolfgang Peti | RRID:Addgene_26566 | |
| Recombinant DNA reagent (plasmid) | pET28 PP1-Phactr1 fusion | *Fedoryshchak et al., 2020* | | |
| Recombinant DNA reagent (plasmid) | pET28 PP1-PIP fusions | This paper | | See Methods |
| Recombinant DNA reagent (plasmid) | pcDNA5 PP1-PIP fusions | This paper | | See Methods |
| Recombinant DNA reagent (plasmid) | pcDNA5/FRT/TO | Thermo V652020 | | |
| Recombinant DNA reagent (plasmid) | pEF mCherry-4E-BP1 (wt/deltaPBM/118+A/S65A/S101A/SS65,101AA) | This paper | | See Methods |
| Recombinant DNA reagent (plasmid) | pOG44 | Thermo V600520 | | |
| Recombinant DNA reagent | pGro7 plasmid | Takara 3340 | | |
| Sequence-based reagent | Oligonucleotides | This paper | | See Methods |
| Sequence-based reagent | Peptides | This paper | | See Methods |
| Commercial assay or kit | Q5 Site-Directed Mutagenesis Kit | NEB e0552s | | |
| Commercial assay or kit | NEBuilder HiFi DNA Assembly Cloning Kit | NEB e5520s | | |
| Commercial assay or kit | TMT10plex Isobaric Label Reagent Set, 0.8 mg | Thermo 90111 | | |
| Commercial assay or kit | High-Select Fe-NTA Phosphopeptide Enrichment Kit | Thermo A32992 | | |
| Commercial assay or kit | High-Select TiO2 Phosphopeptide Enrichment Kit | Thermo A32993 | | |
| Commercial assay or kit | High pH Reversed Phase Fractionation Kit | Pierce 84868 | | |
| Commercial assay or kit | Transcriptor First Strand cDNA Synthesis kit | Roche 04897030001 | | |
| Commercial assay or kit | RFP-trap magnetic agarose beads | ChromoTek rtma-20 | RRID:AB_2827596 | |
| Commercial assay or kit | Phosphate sensor | Thermo PV4406 | | |
| Chemical compound | Lipofectamine 2000 | Invitrogen 11668-019 | | |
| Software | Xcalibur | Thermo | RRID:SCR_014593 | https://www.thermofisher.com/order/catalog/product/OPTON-30965 |

*Appendix 1 Continued on next page*

*Appendix 1 Continued*

| Reagent type (species) or resource | Designation | Source or reference | Identifiers | Additional information |
|---|---|---|---|---|
| Software | MaxQuant | *Cox and Mann, 2008* | RRID:SCR_014485 | https://cox-labs.github.io/coxdocs/maxquant_instructions.html |
| Software | Perseus | *Tyanova et al., 2016* | RRID:SCR_015753 | https://www.maxquant.org/perseus/ |
| Software | Weblogo | University of California, Berkeley | RRID:SCR_010236 | https://weblogo.berkeley.edu/logo.cgi/ |
| Software | GraphPad Prism | GraphPad | RRID:SCR_002798 | https://www.graphpad.com/scientific-software/prism/ |
| Software | Image Studio Lite 5.2 | LI-COR | RRID:SCR_013715 | https://www.licor.com/bio/image-studio-lite/ |
| Software | SnapGene software | Insightful Science | RRID:SCR_015052 | https://www.snapgene.com/ |
| Software | FlowJo v10.10.0 | BD Biosciences | RRID:SCR_008520 | https://flowjo.com/ |
| Software | AlphaFold 3 | Google Deepmind | RRID:SCR_021709 | https://alphafoldserver.com/ |
| Chemical compound | Manganese Chloride | Fluka 221279-500G | | |
| Chemical compound | Arabinose | Biosynth limited MA02043 | | |
| Chemical compound | IPTG | Neo Biotech NB-45-00030-25G | | |
| Chemical compound | Chloramphenicol | Acros organic 227920250 | | |
| Chemical compound | Tris | SDS 10708976001 | | |
| Chemical compound | Imidazole | Sigma-Aldrich I2399-100G | | |
| Chemical compound | Sodium Chloride | Sigma-Aldrich S9888-1KG | | |
| Chemical compound | Triton X-100 | Sigma-Aldrich X100-100ML | | |
| Chemical compound | TCEP | Fluorochem M02624 | | |
| Chemical compound | AEBSF | Melford A20010-5.0 | | |
| Chemical compound | Benzamidine | Melford B4101 | | |
| Chemical compound | Complete EDTA Free Protease Inhibitor tablet | Roche 05056489001 | | |
| Chemical compound | Glutathione Sephanrose 4B | GE Healthcare 17-0756-05 | | |
| Chemical compound | Ni-NTA Agarose | QIAGEN 30230 | | |
| Chemical compound | Tween 20 | Sigma-Aldrich P1379-100ML | | |
| Chemical compound | BSA | Sigma-Aldrich A2153-100G | | |
| Chemical compound | Lithium Chloride | Hampton Research HR2-631 | | |
| Chemical compound | Tri Sodium Citrate | Hampton Research HR2-549 | | |
| Chemical compound | PEG 6000 | Hampton Research HR2-533 | | |
| Chemical compound | PEG 3350 | Hampton Research HR2-527 | | |
| Chemical compound | Sodium Bromide | Hampton Research HR2-699 | | |
| Chemical compound | Potassium citrate | Hampton Research HR2-683 | | |

*Appendix 1 Continued on next page*

*Appendix 1 Continued*

| Reagent type (species) or resource | Designation | Source or reference | Identifiers | Additional information |
|---|---|---|---|---|
| Chemical compound | Bis-Tris-Propane | Sigma-Aldrich B6755-500G | | |
| Chemical compound | Sodium Iodide | Sigma-Aldrich 383112–100G | | |
| Chemical compound | Glycerol | SDS G7893-2L | | |
| Chemical compound | Ethylene Glycol | Sigma-Aldrich 324558–1L | | |
| Chemical compound | BTTAA | Cayman 41089 | | |
| Chemical compound | CuSO4 pentahydrate | Sigma-Aldrich C1297-100g | | |
| Chemical compound | Alexa Fluor-488 azide | Fisher 10033964 | | |
| Chemical compound | Tetracycline hydrochloride | Sigma-Aldrich T7660-25g | | |
| Chemical compound | Zeocin | Invivogen ant-zn-1 | | |
| Chemical compound | Blasticidin | Invivogen ant-bl-1 | | |
| Chemical compound | Hygromycin B | Invivogen ant-hg-2 | | |

