## [Editor Report · eLife Assessment]

This **important** study reports on a basis for neurabin-mediated specification of substrate choice by protein phosphatase-1. The data from the comprehensive approach using structural, biochemical, and computational methods are **compelling**. This paper is broadly relevant to those investigating various cellular signaling cascades that entail phosphorylation as the main mechanism.

---

## [Referee Report · Reviewer #1 (Public review)]

Summary:

In this manuscript the Treisman and colleagues address the question of how protein phosphatase 1 (PP1) regulatory subunits (or PP1-interacting protein (PIPs)) confer specificity on the PP1 catalytic subunit which by itself possesses little substrate specificity. In prior work the authors showed that the PIP Phactrs confers specificity by remodelling a hydrophobic groove immediately adjacent to the PP1 catalytic site through residues within the RVxF- ø ø -R-W string of Phactrs. Specifically, the residues proximal and including the 'W' of the RVxF- ø ø -R-W string remodel the hydrophobic groove. Other residues the of the RVxF- ø ø -R-W string (i.e. the RVxF- ø ø -R) are not involved in this remodelling.

The authors suggest that the RVxF- ø ø -R-W string is a conserved feature of many PIPs including PNUTS, Neurabin/spinophilin and R15A. However from a sequence and structural perspective only the RVxF- ø ø -R- is conserved. The W is not conserved in most and in the R15A structure (PDB:7NZM) the Trp side chain points away from the hydrophobic channel - this could be a questionable interpretation due to model building into the low resolution cryo-EM map (4 A).

In this paper the authors convincingly show that Neurabin confers substrate specificity through interactions of its PDZ domain with the PDZ domain-binding motif (PBM) of 4E-BP. They show the PBM motif is required for Neurabin to increase PP1 activity towards 4E-BP and a synthetic peptide modelled on 4E-BP and also a synthetic peptide based on IRSp53 with a PBM added. The PBM of 4E-BP1 confers high affinity binding to the Neurabin PDZ domain. A crystal structure of a PP1-4E-BP1 fusion with Neurabin shows that the PBM of 4E-BP interacts with the PDZ domain of Neurabin. No interactions of 4E-BP and the catalytic site of PP1 are observed. Cell biology work showed that Neurabin-PP1 regulates the TOR signalling pathway by dephosphorylating 4E-BPs.

Strengths:

This work demonstrates convincingly using a variety of cell biology, proteomics, biophysics and structural biology that the PP1 interacting protein Neurabin confers specificity on PP1 through an interaction of its PDZ domain with a PDZ-binding motif of 4E-BP1 proteins. Remodelling of the hydrophobic groove of the PP1 catalytic subunit is not involved in Neurabin-dependent substrate specificity, in contrast to how Phactrs confers specificity on PP1. The active site of the Neurabin/PP1 complex does not recognise residues in the vicinity of the phospho-residue, thus allowing for multiple phospho-sites on 4E-BP to be dephosphorylated by Neurabin/PP1. This contrasts with substrate specificity conferred by the Phactrs PIP that confers specificity of Phactrs/PP1 towards its substrates in a sequence-specific context by remodelling the hydrophobic groove immediately adjacent to the catalytic. The structural and biochemical insights are used to explore the role of Neurabin/PP1 in dephosphorylation 4E-BPs in vivo, showing that Neurabin/PP1 regulates the TOR signalling pathway, specifically mTORC1-dependent translational control.

Weaknesses:

The only weakness is the suggestion that a conserved RVxF- ø ø -R-W string exists in PIPs. The 'W' is not conserved in sequence and 3-dimensions in most of the PIPs discussed in this manuscript. The lack of conservation of the W would be consistent with the finding based on multiple PP1-PIP structures that apart from Phactrs, no other PIP appears to remodel the PP1 hydrophobic channel.

Comments on revisions:

The authors have addressed my comments.

One aspect of the manuscript and response to reviewers is misleading regarding the statement: 'Like many PIPs, they interact with PP1 using the previously defined "RVxF", "ΦΦ", and "R" motifs (Choy et al, 2014).' This statement, and similar in the authors' response, implies that Choy et al discovered the "RVxF" and "ΦΦ" motifs. The Choy et al, 2014 paper reports the discovery of the "R" motif. The "RVxF" and "ΦΦ" motifs were discovered and reported in earlier papers not cited in the authors' manuscript. Perhaps the authors can correct this.

---

## [Referee Report · Reviewer #2 (Public review)]

This manuscript explores the molecular mechanisms that are involved in substrate recognition by the PP1 phosphatase. The authors previously showed that the PP1 interacting protein (PPI), PhactrI, conferred substrate specificity by remodelling the PP1 hydrophobic substrate groove. In this work, the authors aimed to understand the key determinant of how other PIPs, Neurabin and Spinophilin, mediate substrate recognition.

The authors generated a few PP1-PIP fusion constructs, undertook TMT phosphoproteomics and validated their method using PP1-Phactr1/2/3/4 fusion constructs. Using this method, the authors identified phsophorylation sites controlled by PP1-Neurabin and focussed their work on 4E-BP1, thereby linking PP1-Neurabin to mTORC1 signalling. Upon validating that PP1-Neurabin dephosphorylates 4E-BP1, they determined that 4E-BP1 PBM binds to the PDZ domain of Neurabin with an affinity that was greater than 30 fold as compared to other substrates. PP1-Neurabin dephosphorylated 4E-BP1WT and IRSp53WT with a catalytic efficiency much greater than PP1 alone. However, PP1-Neurabin bound to 4E-BP1 and IRSp53 mutants lacking the Neurabin PDZ domain with a catalytic efficiency lesser than that observed with 4E-BP1WT. These results indicate the involvement of the PDZ domain in facilitating substrate recruitment by PP1-Neurabin. Interestingly, PP1-Phactr1 dephosphorylation of 4E-BP1 phenocopies PP1 alone, while PP1-Phactr1 dephosphorylates IRSp53 to a much higher extent than PP1 alone. These results highlights the importance of the PDZ domain and also shed light on how different PP1-PIP holoenzymes mediate substrate recognition using distinct mechanisms. The authors also show that the remodelling of the hydrophobic PP1 substrate groove which is essential for substrate recognition by PP1-Phactr1, was not required by PP1-Neurabin. Additionally, the authors also resolved the structure of a PP1-4E-BP1 fusion with the PDZ-containing C-terminal of Neurabin and observed that the Neurabin/PP1-4E-BP1 complex structure was oriented at 21{degree sign} to that in the unliganded Spinophilin/PP1 complex (resolved by Ragusa et al., 2010) owing to a slight bend in the C-terminal section that connects it to the RVxF-ΦΦ-R-W string. Since, no interaction was observed with the remodelled PP1-Neurabin hydrophobic groove, the authors utilised AlphaFold3 to further answer this. They observed a high confidence of interaction between the groove and phosphorylated substrate and a low confidence of interaction between the groove and unphosphorylated substrate, thereby suggesting that the hydrophobic groove remodelling is not involved in PP1-Neurabin recognition and dephosphorylation of 4E-BP1.

In this work, the authors provide novel insights into how Neurabin depends on the interaction between its PDZ domain and PBM domains of potential substrates to mediate its recruitment by PP1. Additionally, they uncover a novel PP1-Neurabin substrate, 4E-BP1. They systematically employ phosphoproteomics, biochemical and structural methods to investigate substrate specifity in a robust fashion. Furthermore, the authors also compares the interactions between PP1-Neurabin to 4E-BP1 and IRSp53 (PP1-Phactr1 substrate) with PP1-Phactr1, to showcase the specificity of the mode of action employed by these complexes in mediating substrate specificity. The authors do employ an innovative PP1-PIP fusion strategy previously explored by Oberoi et al., 2016 and the authors themselves in Fedoryshchak et al., 2020. This method, allows for a more controlled investigation of the interactions between PP1-PIPs and its substrates. Furthermore, the authors have substantially characterised the importance of the PDZ domain using their fusion constructs, however, I believe that a further exploration into either structural or AlphaFold3 modelling of PBM domain substrate mutants, or a Neurabin PDZ-domain mutant might further strengthen this claim. Overall, the paper makes a substantial contribution to understanding substrate recognition and specificity in PP1-PIP complexes. The study's innovative methods, biological relevance, and mechanistic insights are strengths, but whether this mechanism occurs in a physiological context is unclear.

---

## [Referee Report · Reviewer #3 (Public review)]

Protein Phosphatase 1 (PP1), a vital member of the PPP superfamily, drives most cellular serine/threonine dephosphorylation. Despite PP1's low intrinsic sequence preference, its substrate specificity is finely tuned by over 200 PP1-interacting proteins (PIPs), which employ short linear motifs (SLIMs) to bind specific PP1 surface regions. By targeting PP1 to cellular sites, modifying substrate grooves, or altering surface electrostatics, PIPs influence substrate specificity. Although many PIP-PP1-substrate interactions remain uncharacterized, the Phactr family of PIPs uniquely imposes sequence specificity at dephosphorylation sites through a conserved "RVxF-ΦΦ-R-W" motif. In Phactr1-PP1, this motif forms a hydrophobic pocket that favors substrates with hydrophobic residues at +4/+5 in acidic contexts (the "LLD motif"), a specificity that endures even in PP1-Phactr1 fusions. Neurabin/Spinophilin remodel PP1's hydrophobic groove in distinct ways, creating unique holoenzyme surfaces, though the impact on substrate specificity remains underexplored. This study investigates Neurabin/Spinophilin specificity via PDZ domain-driven interactions, showing that Neurabin/PP1 specificity is governed more by PDZ domain interactions than by substrate sequence, unlike Phactr1/PP1.

A significant strength of this work is the use of PP1-PIP fusion proteins to effectively model intact PP1•PIP holoenzymes by replicating the interactions that remodel the PP1 interface and confer site-specific substrate specificity. When combined with proteomic analyses to assess phospho-site depletion in mammalian cells, these fusions offer critical insights into holoenzyme specificity, revealing new candidate substrates for Neurabin and Spinophilin. The studies present compelling evidence that the PDZ domain of PP1-Neurabin directs its specificity, with the remodeled PP1 hydrophobic groove interactions having minimal impact. This mechanism is supported by structural analysis of the PP1-4E-BP1 substrate fusion bound to a Neurabin construct, highlighting the 4E-BP1/PDZ interaction. This work delivers crucial insights into PP1-PIP holoenzyme function, combining biochemical, proteomic, and structural approaches. It validates the PP1-PIP fusion protein model as a powerful tool, suggesting it may extend to studying additional holoenzymes. While an extremely useful model, it must be considered unlikely the PP1-PIP fusions fully recapitulate the specificity and regulation of the holoenzyme.

---

## [Author Response]

The following is the authors’ response to the original reviews

Response to the public reviews:

We are very pleased to see these positive reviews of our preprint.

Reviewers 1 and 3 raise issues around PIP-PP1 interactions.

(1) Role of the “RVxF-ΦΦ-R-W string”

Most PIPs interact with the globular PP1 catalytic core through short linear interaction motifs (SLiMs) and Choy et al (PNAS 2014) previously showed that many PIPs interact with PP1 through conserved trio of SLiMs, RVxF-ΦΦ-R, which is also present in the Phactrs.

Previous structural analysis showed the trajectory of the PPP1R15A/B, Neurabin/Spinphilin (PPP1R9A/B), and PNUTS (PPP1R10) PIPs across the PP1 surface encompasses not only the RVxF-ΦΦ-R trio, but also additional sequences C-terminal to it (Chen et al, eLife, 2015). This extended trajectory is maintained in the Phactr1-PP1 complex (Fedoryshchak et al, eLife 2020). Based on structural alignment we proposed the existence of an additional hydrophobic “W” SLiM that interacts with the PP1 residues I133 and Y134.

The extended “RVxF-ΦΦ-R-W” interaction brings sequences C-terminal to the “W” SLiM into the vicinity of the hydrophobic groove that adjoins the PP1 catalytic centre. In the Phactr1/PP1 complex, these sequences remodel the groove, generating a novel pocket that facilitates sequence-specific substrate recognition.

This raises the possibility that sequences C-terminal to the extended “RVxF-ΦΦ-R-W string” in the other complexes also confer sequence-specific substrate recognition, and our study aims to test this hypothesis. Indeed, the hydrophobic groove structures of the Neurabin/Spinophilin/PP1 and Phactr1/PP1 complexes differ significantly (Ragusa et al, 2010; see Fedoryshchak et al 2020, Fig2 FigSupp1).

(2) Orientation of the W side chain

Reviewer 1 points out that in the substrate-bound PP1/PPP1R15A/Actin/eIF2 pre-dephosphorylation complex the W sidechain is inverted with respect to its orientation in PP1-PPP1R15B complex (Yan et al, NSMB 2021). The authors proposed that this may reflect the role of actin in assembly of the quaternary complex. This does not necessarily invalidate the notion that sequences C-terminal to the “W” motif might play a role in actin-independent substrate recognition, and we therefore consider our inclusion of the R15A/B fusions in our analysis to be reasonable.

(3) Conservation of W

The motif ‘W’ does not mandate tryptophan - Phactrs and PPP1R15A/B indeed have W at this position but Neurabin/spinophilin contain VDP, which makes similar interactions. Similarly the “RVxF” motifs in Phactr1, Neurabin/Spinophilin, PPP1R15A/B and PNUTS are LIRF, KIKF, KV(R/T)F and TVTW respectively.

In our revision, we will present comparisons of the differentially remodelled/modified PP1 hydrophobic groove in the various complexes, discuss the different orientations of the tryptophan in the previously published PPP1R15A/PP1 and PPP1R15B/PP1 structures. We will also address the other issues raised by the referees.

**Recommendations for the authors:**

**Reviewer #1 (Recommendations for the authors):**
Comments and suggestions for revisions(1) The authors do not provide strong evidence that the interactions of the 'W' of the RVxF- øø -R-W string with the hydrophobic groove of PP1 is conserved in PIPs. Whereas the RVxF motif is well conserved and validated since its discovery in 1997, as are the øø - (an extension of the RVxF motif), and the 'R', the conservation of the Trp residue in the RVxF-øø-R-W string is not conserved.

We did not mean to imply that the W motif is conserved amongst all PIPs.

Most PIPs interact with the globular PP1 catalytic core through short linear interaction motifs (SLiMs). Choy et al (PNAS 2014) previously showed that many PIPs interact with PP1 through a conserved trio of SLiMs, RVxF-ΦΦ-R, which is also present in the Phactrs.

Previous structural analysis showed that the PPP1R15A/B, Neurabin/Spinophilin (PPP1R9A/B), and PNUTS (PPP1R10) PIPs share a trajectory across the PP1 surface that encompasses not only the RVxF-ΦΦ-R SLIMs, but also additional sequences C-terminal to the R SLIM (Chen et al, eLife, 2015). This trajectory is also shared by the Phactr1-PP1 complex (Fedoryshchak et al, eLife, 2020). Based on this structural alignment we proposed the existence of an additional hydrophobic “W” SLiM that interacts with the PP1 residues I133 and Y134 (See Fedoryshchak et al, 2020, Figure 1 figure supplement 2).

Introduction, paragraph 2 is rewritten to make this clearer.

The sequence and positions of W differ in amino acid type and position relative to the RVxF-øø-R string.

The motif ‘W’ does not mandate tryptophan, it is our name for a common structurally aligned motif: although the Phactrs and PPP1R15A/B indeed have W at this position, Neurabin and spinophilin contain VDP, which nevertheless makes similar interactions. Similarly the _“_RVxF” motifs in Phactr1, Neurabin/Spinophilin, PPP1R15A/B and PNUTS are LIRF, KIKF, KV(R/T)F and TVTW respectively.

In the Discussion the authors state that the hydrophobic groove of PP1 is remodelled by Neurabin. However, details of this are not described or shown in the manuscript.

The shared trajectory determined by the RVxF-øø-R-W string brings the sequences C-terminal to the W SLIM into the vicinity of the PP1 hydrophobic groove. In the Phactr1/PP1 holoenzyme this generates a novel pocket required for substrate recognition (Fedoryshchak et al, 2020). These observations raised the possibility that sequences C-terminal to the “W” motif in the other RVxF-øø-R-W PIPs also play a role in substrate recognition.

Introduction paragraph 3 now cites a new Figure 1-S2, which shows how the hydrophobic groove is remodelled in the various different PIP/PP1 complexes. A revised Figure 1A now indicates the hydrophobic residues defining the hydrophobic groove by grey shading.

(2) To add to the confidence of the structure, the authors should include a 2Fo-Fc simulated annealing omit map, perhaps showing the R and W interactions of the RVxF-øø-R-W string.

This is now included as new Figure 6 Figure supplement 1. Note that in Neurabin, the W motif is VDP, where the valine and proline sidechains interact similarly to the tryptophan (see also new Figure 1-S2G,H).

We also add a new supplementary Figure 6-S1 comparing our PBM-liganded Neurabin PDZ domain with the previously published unliganded structure (Ragusa et al 2010).

(3) Page 16. The authors state that spinophilin remodels the PP1 hydrophobic groove differently from Phactrs. Arguably spinophilin does not remodel the PP1 hydrophobic groove at all. There are no contacts between spinophilin and the PP1 hydrophobic groove in the spinophilin-PP1 structure, correlating with the absence of 'W" in the RVxF-øø-R-W string in spinophilin.

The VDP sequence corresponding to the W motif in spinophilin and neurabin makes analogous contacts to those made by the W in Phactr1 (see Fedoryshchak et al 2020).

Remodelling is meant in the sense of altering the structure of the major groove by bringing new sequences into its vicinity rather than necessarily directly interacting with it. The spinophilin/PP1 and Phactr/PP1 hydrophobic grooves are compared in new Figure 1-S2 (see also Fedoryshchak et al 2020, Figure 2 figure supplement 1)

(4) Page 8. For the cell-based/proteomics-dephosphorylation assay in Figure 2, it isn't clear why there were no dephosphorylation sites detected for the PPP1R15A/B-PP1 fusion (except PPP6R1 S531 for PPP1R15B). One might have expected a correlation with PP1 alone. Does this imply that PPP1R15A/B are inhibiting PP1 catalytic activity? Was the activity tested in vitro?

The R15A/B data are compared to average abundance of all the phosphosites in the dataset, including those of PP1.

We have not tested for a general inhibitory effect of R15A/B on PP1 activity. Many PIPs including R15A/B do occlude one or more of the PP1 substrate groove and therefore generally act as inhibitors of PP1 activity against some potential substrates, while enhancing activities against others.

Other points(4) Figure S1: Colour sequence similarities/identities.

Done

(6) Figures: Structure figures lacked labels:Figure 1A, label PP1, Phactrs etc.

Done

Figure 6, label PP1, Neurabin, previous Neurabin structure (Fig. 6C), hydrophobic groove, PDZ domain, etc.

Done

(7) Statistical analysis. p values should be shown for data in:Figure 5.

To avoid cluttering the Figure, a new sheet, “statistical significance” has been added to Supplementary Table 3, summarizing the analysis.

Figure 1.

Figure amended (now figure 1-S1).

(8) Some inconsistency with labels, eg '34-WT' used in Fig. 5C, whereas '34A-WT' (better) in Methods.

Now changed to 34A etc where used.

(9) Page 6. PPP1R9A/B is not shown in Figure 1A and Figure S1A.

PPP1R9A/B are Neurabin and spinophilin - now clarified in Introduction paragraph 2, Results paragraph 1, Discussion paragraph 1.

(10) Page 7: lines 4, 'site' not 'side'.

Done

(11) Page 9: DTL and CAMSAP3 were found to be dephosphorylated in the PP1-Neurabin/spinophilin screen. Are these PDZ-binding proteins?

Neither DTL nor CAMSAP3 contain C-terminal hydrophobic residues characteristic of classical PBMs. Sentence added in Discussion, paragraph 5

(12) Page 12 and Figure 5 and S5: The synthetic p4E-BP1 and IRSp53WT peptides with PBM should be given more specific names to indicate the presence of the PBM.

We have renamed 4E-BP1^WT^ and IRSp53^WT^ to 4E-BP1^PBM^ and IRSp53^PBM^ respectively, emphasising the inclusion of the wildtype or mutated PBM from 4E-BP1 on these peptides.

Text, Figure 5, and Figure S5 all revised accordingly.

(13) Give PDB code for spinophilin-PP1 complex coordinates shown in Figure 6C.

PDB codes for the various PIP/PP1 complexes now given in new Figure 1-S2 and revised Figure 6C.

**Reviewer #2 (Recommendations for the authors):**
The work undertaken by the authors is extensive and robust, however, I believe that some improvement in the writing and some detailed explanation of certain results sections would help with the presentation of the work and clarity for the readers.(1) The introduction should contain more information about the interaction between PP1 and Neurabin, given that this is the focus of the paper. This would give the reader the necessary background required to follow the paper.

Introduction paragraph 2 revised to describe the different SLIMs in more detail. New Figure 1-S2 shows detail of the different remodelled hydrophobic grooves in the various PIP/PP1 complexes.

(2) More information on PP1-IRSp53L460A has to be added before discussing results in S1B.

Sentence explaining that IRSp53 L460 docks with the remodelled PP1 hydrophobic groove in the Phactr1/PP1 holoenzyme added in Results paragraph 2.

(3) Page 6: "as expected, the +5 residue L460A mutation, which impairs dephosphorylation by the intact Phactr1/PP1 holoenzyme, impaired sensitivity to all the fusions, indicating that they recognise phosphorylated IRSp53 in a similar way (Figure S1B)". Statistics between IRSp53 and IRSp53L460A across PP1-PIPs need to be conducted before concluding the above. From the graph and the images, the impairment to dephosphorylation is not convincing.

For each of the four PP1-Phactr fusions, the IRSp53 L460A peptide shows significantly less reactivity than the IRSp53WT peptide (p<0.05 for each fusion).

Since the proteomics studes in Figure 2 show that the substrate specificity of the four PP1-Phactr1 fusions is virtually identical, we combined the data for the four different fusions. The IRSp53 L460A peptide shows significantly less reactivity than the IRSp53WT peptide in this analysis (p< 0.0001). This result shown in revised Figure S1B and legend.

(4) mCherry-4E-BP1(118+A), in which an additional C-terminal alanine should still allow TOSmediated phosphorylation, but prevent PDZ interaction. Does 4EBP1 (118+A) actually prevent interaction between PP1-Neurabin? This interaction needs to be validated, especially since spinophilin was shown to bind to multiple regions of PP1.

It is not clear what the referee is asking for here. The biochemical analysis in Figure 4C shows that the C-terminus of 4E-BP1 constitutes a classical PBM. The X-ray crystallography in Figure 6 confirms this, demonstrating H-bond interactions between the 4E-BP1 C-terminal carboxylate and main chain amides of L514, G515 and I516.

We consider the possibility that the 4E-BP1(118+A) mutant inhibits the activity of PP1-neurabin via a mechanism other than direct blocking 4E-BP1 / PDZ interaction to be unlikely for the following reasons:

(1) Addition of a C-terminal alanine will disrupt the PBM interaction because the extra residue sterically blocks access to the PBM-binding groove. This is the most parsimonious explanation, and is based on our solid structural and biochemical evidence that the 4E-BP1 C-terminus is a classical PBM.

(2) Alphafold3 modelling predicts Neurabin PDZ / 4E-BP1 PBM interaction with high confidence (shown in Figure 6-S2E), but it does not predict any PDZ interaction with 4E-BP1(118+A). Note added in Figure 6-S2 legend.

(3) Recognition of the 4E-BP1(118+A) mutation without loss of binding affinity would require that the mutant becapable of binding formally equivalent to recognition of an “internal” PDZ-binding peptide. Recognition of such “internal peptides” is dependent on their adopting a specifically constrained conformation, which typically requires reorganisation of the PDZ carboxylate-binding GLGF loop. Such “internal site” recognition typically involves more than one residue C-terminal to the conventional PDZ “0” position (see Penkert et al NSMB 2004, doi:10.1038/nsmb839; Gee et al JBC 1998, DOI: 10.1074/jbc.273.34.21980; Hillier et al 1999, Science PMID: 10221915).

(5) It is nice to see that the various PP1-Phactr fusions have around 60% substrate overlap between them. Would it be possible to compare these results with previously published mass spec data of Phactr1XXX from the group? There is mention of some substrates being picked up, but a comparison much like in Figure 2E would be more informative about the extent to which the described method captures relevant information.

This is difficult to do directly as the PP1-Phactr fusion data are from human cells while that in Fedoryshchak et al 2020 is from mouse.

However, manual curation shows that of the 28 top hits seen in our previous analysis of Phactr1XXX in NIH3T3 cells, 18 were also detectable in the HEK293 system; of these, 13 were also detected as as PP1-Phactr fusion hits. Data summarised in new Figure 2-S1C. Text amended in Results, “Proteomic analysis...”, paragraph 2.

(6) Figure 3D Why are the levels of pT70, pT37/46 and total protein in vector controls much lower as compared to 0nM Tet in PP1-Neurabin conditions? It is also weird that given total protein is so low, why are the pS65/101 levels high compared to the rest?

We think it likely these phenomena reflect a low level expression of PP1-Neurabin expression in uninduced cells. Now noted in Figure 3D legend, basal PP1-Neurabin expression shown in new Figure 3-S1C. This alters the relative levels of the different species detected by the total 4E-BP1 antibody in favour of the faster migrating forms, which are less phosphorylated than the slower ones, and the total amount increases about 2-fold (Figure 3D, compare 0nM Tet lanes).

The altered p65/101-pT70 ratio is also likely to reflect the leaky PP1-Neurabin expression, since the relative intensities of the various phosphorylated species are dependent on both the relative rates of phosphorylation and dephosphorylation. Expression of a phosphatase would therefore be expected to differentially affect the phosphorlyation levels of different sites according to their reactivity.

(7) Figure 3E: Does inhibiting mTORC further reduce translation when PP1-Neurabin is expressed? If this is the case, this might suggest that they might not necessarily be mTORC inhibitors?

We have not done this experiment. Since Rapamycin cannot be guaranteed to completely block 4E-BP1 phosphorylation, and PP1-Neurabin cannot be guaranteed to completely dephosphorylate 4E-BP1, any further reduction upon their combination would be hard to interpret.

(8) Substrate interactions with the remodelled PP1 hydrophobic groove do not affect PP1-Neurabin specificity. Is there evidence that PP1-Neurabin remodels the hydrophobic groove? Is it not possible that Neurabin does not remodel the PP1 groove to begin with and hence there is no effect observed with the various mutants? If this is not the case, it should be explained in a bit more detail.

Comparison of the Neurabin/PP1 and Phactr1/PP1 structures shows that the hydrophobic groove is remodelled differently in the two complexes. Now shown in new Figure 1-S2B,C,G.

(9) Figure 5B has a lot of interesting information, which I believe has not been discussed at all in the results section.

To help interpretation of the enzymology in Figure 5 we have renamed 4E-BP1WT and IRSp53WT to 4E-BP1PBM and IRSp53PBM respectively, emphasising the inclusion of the wildtype or mutated PBM from 4E-BP1 on these peptides. Text in Results, “PDZ domain interaction…”, paragraph 1, and Figures 5 and S5 revised accordingly.

Why does the 4E-BP1Mut affect catalytic efficiency of PP1 alone when compared with WT, while no difference is observed with IRSp53WT and mutant?

We do not understand the basis for the differential reactivity of 4E-BP1PBM and 4E-BP1MUT with PP1 alone; we suspect that it reflects the hydrophobicity change resulting from the MDI -> SGS substitution. However this is unlikely to be biologically significant as PP1 is sequestered in PIP-PP1 complexes.

Importantly, the two PP1 fusion proteins behave consistently in this assay – the presence of the intact PBM increases reactivity with PP1-Neurabin, but has no effect on dephosphorylation by PP1-Phactr1.

Why does PP1 alone not have a difference between IRSp53WT and mutant, while PP1-Neurabin does have a difference?

This is due to the presence of the PBM in IRSp53WT (now renamed IRSp53PBM), which affects increases affinity for PP1 Neurabin, but not PP1 alone. Likewise, PP1-Phactr1, which does not possess a PDZ domain, is also unaffected by the integrity of the PBM.

(7) “Strikingly, alanine substitutions at +1 and +2 in 4E-BP1WT increased catalytic efficiency by both fusions, perhaps reflecting changes at the catalytic site itself (Figure 5E, Figure S5E)”. This could be expanded upon, because this suggests a mechanism that makes the substrate refractory to PDZ/hydrophobic groove remodelling?

We favour the idea that this reflects a requirement to balance dephosphorylation rates between the multiple 4E-BP1 phosphorylation sites, especially if multiple rounds of dephosphorylation occur for each PBM—PDZ interaction. Additional sentences added in Discussion paragraph 7.

(8) Typographical errors and minor comments:a) PIPs can target PP1 to specific subcellular locations, and control substrate specificity through autonomous substrate-binding domains, occupation or extension of the substrate grooves, or modification of PP1 surface electrostatics.b) Phosphophorylation side site abundances within triplicate samples from the same cell line were comparable between replicates (Figure 2B).c) While the alanine substitutions had little effect, conversion of +4 to +6 to the IRSp534E-BP1 sequence LLD increased catalytic efficiency some 20-fold (Figure 5C, Figure S5C).d) Figure 3E labels are not clear. The graph can be widened to make the labels of the conditions clearer.

All corrected

**Reviewer #3 (Recommendations for the authors):**
This was a very well-written manuscript.However, I was looking for a summary mechanistic figure or cartoon to help me navigate the results.I noted a few typos in the text.

New summary Figure 5-S2 added, cited in results, and discussed in Discussion paragraph 6,7.